# Randomised trial of coconut oil, olive oil or butter on blood lipids and other cardiovascular risk factors in healthy men and women

Kay-Tee Khaw,[1] Stephen J Sharp,[2] Leila Finikarides,[3,4] Islam Afzal,[5] Marleen Lentjes,[1] Robert Luben,[1] Nita G Forouhi[2]

[1]Department of Public Health and Primary Care, University of Cambridge School of Clinical Medicine, Cambridge, UK
[2]Medical Research Council Epidemiology Unit, University of Cambridge School of Clinical Medicine, Cambridge, UK
[3]BBC Television 'Trust Me I'm a Doctor', BBC Glasgow, Glasgow, UK
[4]Winton Centre for Evidence Communication, University of Cambridge, Cambridge, UK
[5]Aston Medical Research Institute, Aston Medical School, Aston University, Birmingham, UK

**Correspondence to**
Professor Kay-Tee Khaw;
kk101@medschl.cam.ac.uk and Professor Nita G Forouhi;
nita.forouhi@mrc-epid.cam.ac.uk

## ABSTRACT

**Introduction** High dietary saturated fat intake is associated with higher blood concentrations of low-density lipoprotein cholesterol (LDL-C), an established risk factor for coronary heart disease. However, there is increasing interest in whether various dietary oils or fats with different fatty acid profiles such as extra virgin coconut oil may have different metabolic effects but trials have reported inconsistent results. We aimed to compare changes in blood lipid profile, weight, fat distribution and metabolic markers after four weeks consumption of 50 g daily of one of three different dietary fats, extra virgin coconut oil, butter or extra virgin olive oil, in healthy men and women in the general population.

**Design** Randomised clinical trial conducted over June and July 2017.

**Setting** General community in Cambridgeshire, UK.

**Participants** Volunteer adults were recruited by the British Broadcasting Corporation through their websites. Eligibility criteria were men and women aged 50–75 years, with no known history of cancer, cardiovascular disease or diabetes, not on lipid lowering medication, no contraindications to a high-fat diet and willingness to be randomised to consume one of the three dietary fats for 4 weeks. Of 160 individuals initially expressing an interest and assessed for eligibility, 96 were randomised to one of three interventions; 2 individuals subsequently withdrew and 94 men and women attended a baseline assessment. Their mean age was 60 years, 67% were women and 98% were European Caucasian. Of these, 91 men and women attended a follow-up assessment 4 weeks later.

**Intervention** Participants were randomised to extra virgin coconut oil, extra virgin olive oil or unsalted butter and asked to consume 50 g daily of one of these fats for 4 weeks, which they could incorporate into their usual diet or consume as a supplement.

**Main outcomes and measures** The primary outcome was change in serum LDL-C; secondary outcomes were change in total and high-density lipoprotein cholesterol (TC and HDL-C), TC/HDL-C ratio and non-HDL-C; change in weight, body mass index (BMI), waist circumference, per cent body fat, systolic and diastolic blood pressure, fasting plasma glucose and C reactive protein.

**Results** LDL-C concentrations were significantly increased on butter compared with coconut oil (+0.42, 95% CI 0.19 to 0.65 mmol/L, P<0.0001) and with olive oil (+0.38, 95% CI 0.16 to 0.60 mmol/L, P<0.0001), with no differences in change of LDL-C in coconut oil compared with olive oil (−0.04, 95% CI −0.27 to 0.19 mmol/L, P=0.74). Coconut oil significantly increased HDL-C compared with butter (+0.18, 95% CI 0.06 to 0.30 mmol/L) or olive oil (+0.16, 95% CI 0.03 to 0.28 mmol/L). Butter significantly increased TC/HDL-C ratio and non-HDL-C compared with coconut oil but coconut oil did not significantly differ from olive oil for TC/HDL-C and non-HDL-C. There were no significant differences in changes in weight, BMI, central adiposity, fasting blood glucose, systolic or diastolic blood pressure among any of the three intervention groups.

**Conclusions and relevance** Two different dietary fats (butter and coconut oil) which are predominantly saturated fats, appear to have different effects on blood lipids compared with olive oil, a predominantly monounsaturated fat with coconut oil more comparable to olive oil with respect to LDL-C. The effects of different dietary fats on lipid profiles, metabolic markers and health outcomes may vary not just according to the general classification of their main component fatty acids as saturated or unsaturated but possibly according to different profiles in individual fatty acids, processing methods as well as the foods in which they are consumed or dietary patterns. These findings do not alter current dietary recommendations to reduce saturated fat intake in general but highlight the need for further elucidation of the more nuanced relationships between different dietary fats and health.

**Trial registration number** NCT03105947; Results.

## Strengths and limitations of this study

► The randomised trial design comparing three dietary fat interventions minimised confounding and bias.
► There was good compliance and participants were from the general community in a 'real life' setting.
► Objective measures of outcome—blood biochemistry and anthropometry—were used, minimising bias.
► Participants were not blinded as to the intervention, and the intervention was relatively short term over 4 weeks.

## INTRODUCTION

This trial was conducted in the context of debate over longstanding dietary recommendations to reduce dietary fat intake for health. The Women's Health Initiative reported no differences in cardiovascular disease in women randomised to low fat and usual diets over 8 years[1] while an intervention comparing a low-fat diet with a Mediterranean diet with extra virgin olive oil or nuts (PREDIMED) reported approximately 30% lower cardiovascular events in both Mediterranean diet arms after 4.8 years[2]; meta-analyses of observational studies and trials report inconsistent findings in the relationship between dietary saturated fatty acids and cardiovascular disease[3 4] and the relationships of dairy fats including milk and butter with cardiovascular disease also being debated.[5–7] Part of the debate relates to the increasing evidence that different individual fatty acids, such as the odd chain or even chain saturated fatty acids, or short, medium and long chain saturated fatty acids, may have different metabolic pathways and subsequent potential health effects as well as the understanding that diet is more complex than individual nutrients or generic biochemical nutrient groups and that contextual factors such as foods and dietary patterns are important. The 2015–2020 US dietary guidelines[8] now focus on foods and dietary patterns and while they recommend limiting saturated and trans fats, they no longer explicitly recommend limiting total fat. In this context therefore, there is renewed interest in the health effects of different fats and oils.

Extra virgin coconut oil has recently been promoted as a healthy oil. Though high in saturated fat, the main saturated fatty acid, lauric acid (c12:0), has been suggested to have different metabolic and hence health effects compared with other saturated fatty acids such as palmitic acid (c16:0), predominant in butter, palm oil and animal fat. In particular, it has been suggested that coconut oil does not raise total cholesterol (TC) or low-density lipoprotein cholesterol (LDL-C) as much as butter. A recent review on coconut oil and cardiovascular risk factors in humans concluded that the evidence of an association between coconut oil consumption and blood lipids or cardiovascular risk was mostly poor quality.[9] While some small studies have been reported comparing coconut oil and butter, these have been small[10 11] and none conducted in the UK where overall dietary patterns are different from Asia, USA or New Zealand where most trials have been conducted. The 2017 American Heart Association Presidential advisory on dietary fats and cardiovascular disease highlighted the paucity of evidence over the long-term health effects of saturated fats such as coconut oil and reinforced strongly recommendations to lower dietary saturated fat and replacement with unsaturated fat to lower LDL-C and prevent cardiovascular disease.[12] In particular, they stated 'because coconut oil increases LDL-C, a cause of cardiovascular disease and has no known off setting favourable effects, we advise against the use of coconut oil'.[12]

Though the PREDIMED study reported lower cardiovascular disease events in those randomised to extra virgin olive oil or added nuts,[2] this trial reported no overall effects on LDL-C or TC for those on olive oil compared with the low-fat diet,[13] results consistent with a review of intervention trials of high phenolic olive oil.[14]

We therefore aimed to examine whether in free living healthy men and women in the UK, we could observe differences in blood lipids after 1 month's consumption of 50 g daily of one of three different fats within the context of their usual diet. Although this was a short-term trial that did not address cardiovascular disease events, blood lipids are a well established risk factor for coronary heart disease and the aim was to compare directly the effects of three different fats, extra virgin coconut oil, butter (both predominantly saturated fats) with extra virgin olive oil (monounsaturated fat) on blood lipid profiles and metabolic measures, in a pragmatic trial using amounts feasible in daily diets.

## METHODS

### Study population

Participants were volunteers living in the general community predominantly in the Cambridgeshire area, recruited through British Broadcasting Corporation (BBC) advertising in May and June 2017. Eligible participants were men or women aged between 50 and 75 years who did not have a known medical history of heart disease, stroke, cancer or diabetes and who were not taking medication for lowering blood lipids such as statins. They had to be willing to be randomised to consume 50 g daily of one of the designated fats for four weeks and not have any contraindications to eating a high-fat diet such as gall bladder or bowel problems. Of 160 individuals expressing an interest, 96 were eligible and randomised to the intervention, 2 withdrew prior to the start of the study and 94 attended a baseline assessment.

### Allocation to intervention

Participants were assigned a unique study identification number (ID). These ID numbers were randomised by computer generated allocation conducted by an independent statistician separately in men and women, into one of three parallel intervention arms approximately equal in size: extra virgin coconut oil, butter or extra virgin olive oil.

### Intervention

Participants attending the baseline assessment, at the end of their appointment, received 1 month's supply of one of the three different dietary fats to which they had been randomly allocated: extra virgin coconut oil or butter or extra virgin olive oil. The BBC study organiser was given an ID list with the random allocation to the fats/oils and was responsible for giving each participant their supply of fat/oils. They were asked to eat 50 g of these fats daily for 4 weeks and given measuring cups for the

50 mL fat and oils: butter was prepacked in 20 g and 30 g portions. They were asked to continue with their usual diet and either incorporate the fat or oil into their daily diet to substitute for other fats or oils or they could eat these fats as a supplement. They also had information sheets with suggestions for how the fats could be consumed including recipes. The fats selected were standard products available from supermarkets bought from suppliers; organic extra virgin coconut oil, organic unfiltered extra virgin olive oil and organic unsalted butter. Samples of the oils/fats used in the trial were sent to a reference laboratory: the West Yorkshire Analytic Services, a UKAS accredited testing service for food composition.

## Assessments

Participants attended two assessments at a community centre in Cambridge: one at baseline before the start of the intervention in June 2017 and one at the end of 4 weeks in July 2017. Prior to their initial assessment, they were asked to fill in a short questionnaire about their health and lifestyle including physical activity and diet as well as complete an online 24 hours dietary assessment questionnaire with automated nutrient intake estimation, developed in Oxford, the DietWebQ.[15] All assessments were conducted between 08:00 and 12:30 hours. Participants were all fasted for a minimum of 4 hours prior to attending the assessment; the majority were fasted overnight. They had height and waist circumference measured to a standardised protocol in light clothing without shoes and blood pressure measured using an automated OMRON device after being seated resting for 5 min. The mean of two readings for blood pressure, height and waist was used for analysis. Weight and per cent body fat were measured using a Tanita body composition monitor. All measurements were conducted by two trained observers unaware of allocation to the oils/fats. Participants gave a 20 mL blood sample which was stored in a 4°C refrigerator and then sent to the laboratory by courier for same day sample processing and storage for later analysis.

After 4 weeks at the end of the intervention, they attended again for a follow-up assessment where the same measurements of height, waist circumference, blood pressure, weight and per cent body fat were conducted, and another fasting 20 mL blood sample taken. Measurements were recorded on new forms and observers and participants did not have access to the measurements taken at the baseline visit. Just prior to this visit, participants were asked to fill in again the online 24-hour DietWebQ. Participants also filled in short questionnaire about their experiences on the intervention fats. This included a question about their overall experience of consuming the assigned oil/fat in the study where they were asked on average, over the past 4 weeks whether they felt mostly the same as usual, mostly felt better than usual or mostly felt worse than usual with an open-ended section for comments including side effects and overall compliance with consuming the fats which they were

asked to self-rate between 0% and 100%. They were also asked whether they changed their type, level or frequency of physical activity in the past month since being in the study and had three options, no overall change in activity, increase in activity or decrease in activity.

Blood samples were identified only by a study ID number and were processed using standard protocols and assayed in two batches at the end of the baseline and follow-up assessments in the Core Biochemical Assay Laboratory (CBAL) Cambridge University Hospitals which has UKAS Clinical Pathology Accreditation; blood samples from individuals on different interventions were thus all assayed in the same batch. The laboratory assays were conducted in a blinded fashion without any indication of the allocated intervention. Cholesterol (TC) and triglycerides were measured using enzymatic assays,[16 17] high-density lipoprotein cholesterol (HDL-C) was measured using a homogenous accelerator selective detergent assay automated on the Siemens Dimension RxL analyser and LDL-C was calculated from the triglyceride, HDL and cholesterol concentrations as described in the Friedewald formula [LDL = Cholesterol - HDL – (Triglycerides/2.2)].[18] Total to HDL-C ratio was computed, and non-HDL-C was computed as TC minus HDL-C.

Plasma glucose was measured using the hexokinase-glucose-6-phosphate dehydrogenase method, and high-sensitivity human C reactive protein was assayed using an automated colourimetric immunoassay: Siemens Dimension CCRP *Cardio*Phase high-sensitivity CRP.

## Trial outcomes

The trial was registered in April 2017 with clinical trials registration: NCT03105947. The primary outcome of the trial was change in LDL-C from baseline to follow-up. Secondary outcomes were change in each of the following variables from baseline to follow-up: TC, HDL-C, triglycerides; ratio of TC/HDL-C, non-HDL-C, fasting blood glucose, C reactive protein, weight, body mass index (BMI), body fat %, waist circumference, systolic blood pressure and diastolic blood pressure.

## Statistical analysis

The study aimed to recruit a total of 90 participants: 30 individuals per group would provide approximately 80% power to detect a difference in mean within-person change in LDL-C (baseline to follow-up) comparing pairs of randomised groups (butter vs coconut oil and butter vs olive oil) of approximately 0.5 mmol/L, assuming a SD of LDL-C of 1.04 mmol/L[19] and a correlation between baseline and follow-up values of 0.79[20] incorporated using the method described by Borm *et al.*[21] With 2 primary pairwise comparisons, the significance level for each comparison was set to 2.5%.

This magnitude of difference was what can be estimated from metabolic ward studies in which replacement of 10% dietary calories from saturated fat is associated with 0.52 mmol/L cholesterol difference[22] though

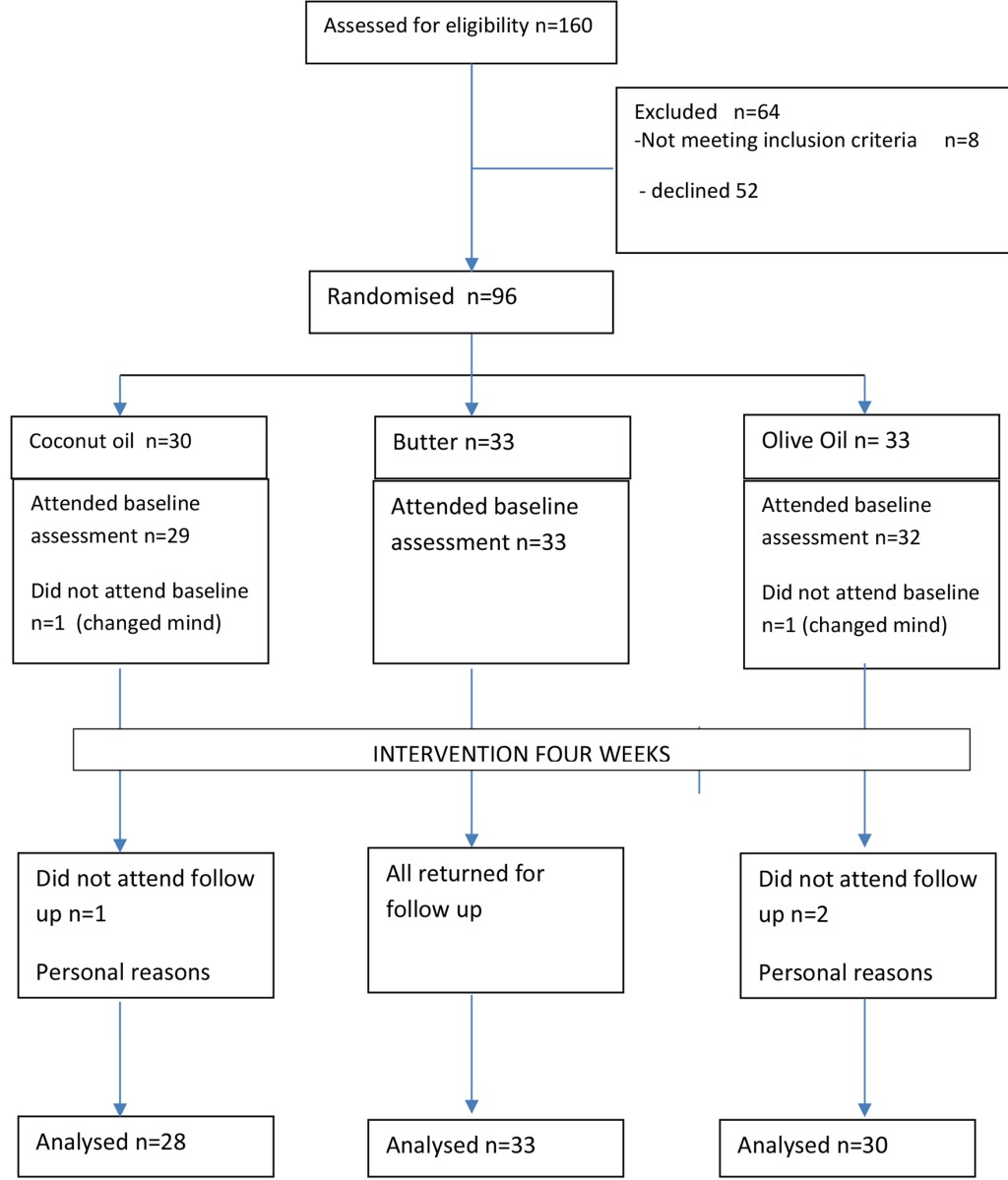

**Figure 1** Recruitment and flow diagram (CONSORT) for coconut oil, olive oil or butter trial.

this did not specify the food sources of saturated fats, and a small intervention trial (n=28) comparing butter and coconut oil with sunflower oil.[10]

Baseline characteristics were summarised separately for each randomised group. As recommended by CONSORT, no P values were calculated for this table. The primary analysis used an intention-to-treat (ITT) population, which included all individuals in the group to which they were randomised, regardless of the extent to which they adhered to the intervention. A secondary analysis used a per protocol (PP) population. This was a subset of the ITT population consisting of those individuals who adhered to the intervention. Participants who reported >75% adherence when asked at the follow-up visit were included in the PP population.

For each outcome, a P value was calculated to compare the three randomised groups using a linear regression model, in which change from baseline was the outcome and including a dummy variable for randomised group and the baseline value of the outcome variable as covariates, that is, an analysis of covariance model. Differences between each pair of randomised groups and 95% CIs were also estimated from a similar model.

### Patient and public involvement

The BBC originally proposed the idea of a study to examine claims about the health benefits of coconut oil in response to public interest; the study would be part of their 'Trust me, I'm a doctor' series. The study was designed as a randomised trial with participants from the general community in discussion with the BBC.

### RESULTS

Figure 1 is the CONSORT diagram for the trial. The recruitment was conducted by the BBC coordinator

through BBC website advertising. From 160 individuals initially expressing an interest and after exclusion criteria, 96 individuals were randomised and invited to a baseline assessment session in June 2017. Two individuals subsequently withdrew and 94 individuals attended the baseline assessment session in June 2017. At the 4-weekfollow-up assessment in July 2017, 91 individuals attended; three individuals did not attend follow-up indicating personal circumstances.

Table 1 shows descriptive characteristics for the participants at the baseline assessment according to the allocation to dietary oils/fats. Two thirds of the participants were women and the mean age overall was 60 years.

Table 2 shows mean changes in the primary and secondary outcomes at the 4-week follow-up within each randomised group and comparisons between each pair of randomised groups. LDL-C concentrations were significantly increased on butter compared with coconut oil (+0.42, 95% CI 0.19 to 0.65 mmol/L, P<0.0001) and olive oil (+0.38 L, 95% CI 0.16 to 0.60 mmol/L, P<0.0001), with no differences in change of LDL-C in coconut oil compared with olive oil (−0.04, 95% CI −0.27 to 0.19 mmol/L, P=0.74). Coconut oil significantly increased HDL-C compared to butter (+0.18 95% CI 0.06 to 0.30 mmol/L) or olive oil (+0.16, 95% CI 0.03 to 0.28 mmol/L).

Butter significantly increased the cholesterol/HDL-C ratio compared with coconut oil (+0.36, 95% CI 0.18 to 0.54) and olive oil (+0.22, 95% CI 0.04 to 0.40) and also increased non-HDL-C compared with coconut oil (+0.39, 95% CI 0.16 to 0.62 mmol/L) and olive oil (+0.39 (95% CI 0.16 to 0.62) but coconut oil did not significantly differ from olive oil for change in cholesterol/HDL-C ratio (−0.14, 95% CI −0.33 to 0.05) or non-HDL-C (0.002, 95% CI −0.23 to 0.24 mmol/L).

Coconut oil also significantly lowered C reactive protein in comparison with olive oil (−0.58, 95% CI −1.12 to −0.04 mg/L) but not compared with butter. There were no significant differences in changes in weight, BMI, central adiposity, fasting blood glucose, systolic or diastolic blood pressure among any of the three intervention groups. For weight, for example, the estimated mean (SD) changes in weight were +0.27 (0.77) kg, 0.04 (1.00) kg and −0.04 (0.84) kg for coconut oil, butter and olive oil, respectively. Adjusting for age, sex and body mass index did not materially alter the results (online supplementary table 1).

Figure 2 shows the difference in the primary outcome (LDL-C) between each pair of randomised groups in the 91 individuals who attended baseline and follow-up. Figures 3–5 show the differences in secondary outcomes comparing butter versus coconut oil, coconut oil versus olive oil and butter versus olive oil, respectively. For comparability, the differences are reported in units of baseline SD for the different outcomes in figures 3–5.

Self-reported compliance was high: 87% of participants reported more than 75% compliance with the intervention over the 4 weeks which was similar among the groups (86% coconut oil, 88% butter and 85% olive oil). Secondary analyses on the 82 participants reporting more than 75% compliance showed similar results (not shown). Reported experience consuming the fats was similar between groups: 57%, 66% and 60% reported feeling no different, 18%, 6% and 13% reported feeling better and 25%, 27% and 23% reported feeling worse in the coconut oil, butter and olive oil groups, respectively. Comparison of dietary intake using the 24-hour DietWebQ showed similar levels of dietary intake across intervention groups at baseline. Following the intervention, total fat intake increased in all intervention groups but estimates for absolute intakes of carbohydrate, protein and alcohol did not differ between intervention groups (table 3). Most of the participants reported no changes in usual physical activity (79%, 73% and 89% no change; 14%, 15% and 4% increased usual physical activity and 7%, 12% and 7% decreased usual physical activity in the coconut oil, butter and olive oil groups, respectively). In a posthoc exploratory analysis, exclusion of individuals who reported increasing usual physical activity had little effect on significant differences between interventions for LDL-C and HDL-C and did not alter the findings for weight change (online supplementary table 2).

Online supplementary appendix 1 shows the fatty acid composition of the three oils/fats used in the intervention. Coconut oil was 94% saturated fatty acids, of which the main components were lauric acid C12:0 (48%), myristic acid C14:0 (19%) and palmitic acid C16:0 (9%). Butter was 66% saturated fatty acids, of which the main components were palmitic acid C16:0 (28%), stearic acid C18:0 (12%) and myristic acid C14:0 (11%). Olive oil was 19% saturated fatty acids, mainly palmitic acid C16 (15%) with stearic acid C18:0 (3%) and 68% monounsaturates with the main component being oleic acid C18:1n9 (64%). These profiles are very similar to those reported from other studies.[9]

## DISCUSSION

In this trial, middle-aged men and women living in the general community were randomly allocated to consume 50 g extra virgin coconut oil or 50 g butter or 50 g extra virgin olive oil for 4 weeks. We observed at the end of the trial significantly different changes in LDL-C and HDL-C concentrations between the three intervention groups; in pairwise comparisons, coconut oil did not significantly raise LDL-C concentrations compared with olive oil while butter significantly raised LDL-C concentrations compared with both coconut oil and olive oil. Coconut oil significantly raised HDL-C concentrations compared with both butter and olive oil. Butter also significantly raised cholesterol/HDL-C ratio and non-HDL-C more than both coconut oil and olive oil but there were no differences between coconut oil and olive oil for changes in cholesterol/HDL-C and non-HDL-C.

There were no significant differences in weight or BMI change, change in central adiposity as measured

**Table 1** Descriptive characteristics at baseline assessment of participants in the COB trial according to allocation (intention to treat)

| | Coconut oil | Butter | Olive oil |
|---|---|---|---|
| | n=29 | n=33 | n=32 |
| | **Mean (SD)** | **Mean (SD)** | **Mean (SD)** |
| Age (years) | 59.1 (6.1) | 61.5 (5.8) | 59.1 (6.4) |
| LDL-cholesterol (mmol/L) | 3.5 (0.9) | 3.5 (0.9) | 3.7 (1.0) |
| Total cholesterol (mmol/L) | 5.9 (1.0) | 5.9 (1.0) | 6.0 (0.9) |
| HDL-cholesterol (mmol/L) | 2.0 (0.5) | 1.9 (0.5) | 1.8 (0.5) |
| Cholesterol/HDL ratio | 3.2 (0.9) | 3.2 (0.8) | 3.5 (1.2) |
| Non-HDL-cholesterol (mmol/L) | 3.9 (1.0) | 4.0 (0.9) | 4.2 (1.1) |
| Glucose (mmol/L) | 5.3 (0.4) | 5.4 (0.5) | 5.4 (0.5) |
| Weight (kg) | 73.9 (15.1) | 70.8 (11.7) | 71.1 (14.5) |
| Waist (cm) | 85.4 (11.9) | 83.7 (8.1) | 86.2 (11.5) |
| Body fat (%) | 29.7 (10.2) | 29.2 (9.0) | 31.5 (9.6) |
| Body mass index (kg/m$^2$) | 25.5 (4.5) | 24.8 (3.5) | 25.0 (4.5) |
| Systolic blood pressure (mm Hg) | 131.4 (18.8) | 136.5 (18.8) | 133.1 (16.5) |
| Diastolic blood pressure (mm Hg) | 79.8 (9.3) | 81.0 (12.0) | 78.1 (6.7) |
| DietWebQ intake/day | | | |
| Total energy (MJ) | 9.00 (3.70) | 8.23 (2.17) | 9.51 (3.5) |
| Protein % energy | 14.8 (4.4) | 16.0 (3.7) | 15.7 (3.0) |
| Carbohydrate % energy | 43.6 (8.9) | 41.4 (8.7) | 42.7 (11.7) |
| Total fat % energy | 37.3 (7.3) | 36.7 (8.7) | 36.4 (10.3) |
| Saturated fat % energy | 14.1 (3.6) | 13.3 (4.4) | 13.4 (4.9) |
| Alcohol % energy | 4.2 (5.4) | 5.9 (7.5) | 5.1 (6.1) |
| Hours of walking in past week | 8.9 (9.5) | 10.9 (12.3) | 10.1 (8.7) |
| Hours of cycling in past week | 1.8 (2.6) | 2.0 (2.5) | 2.7 (5.5) |
| Hours of other physical exercise in past week | 3.4 (3.4) | 2.3 (4.0) | 1.8 (2.6) |
| | n=29 | n=33 | n=32 |
| | **Median (IQR)** | **Median (IQR)** | **Median (IQR)** |
| Triglycerides (mmol/L) | 0.89 (0.74 to 1.10) | 0.92 (0.70 to 1.20) | 0.94 (0.79 to 1.31) |
| C reactive protein (mg/L) | 1.04 (0.47 to 2.15) | 1.08 (0.64 to 2.13) | 1.13 (0.58 to 2.67) |
| | **% (N)** | **% (N)** | **% (N)** |
| Sex | | | |
| Men | 37.9 (11) | 33.3 (11) | 28.1 (9) |
| Women | 62.1 (18) | 66.7 (22) | 71.9 (23) |
| Ethnicity | | | |
| White | 96.6 (28) | 97.0 (32) | 93.8 (30) |
| Non-white | 3.4 (1) | 3.0 (1) | 3.1 (1) |
| Smoking status | | | |
| Never | 58.6 (17) | 66.7 (22) | 68.8 (22) |
| Former | 34.5 (10) | 33.3 (11) | 25.0 (8) |
| Current | 6.9 (2) | 0.0 (0) | 6.3 (2) |
| Alcohol consumption in past year | | | |
| Never or once per month | 20.7 (6) | 30.3 (10) | 28.1 (9) |
| 1–4 times per week | 72.4 (21) | 48.5 (16) | 59.4 (19) |
| Almost every day or every day | 6.9 (2) | 21.2 (7) | 12.5 (4) |
| Highest level of education | | | |
| School to age 16 | 13.8 (4) | 12.1 (4) | 15.6 (5) |
| School to age 18 | 27.6 (8) | 9.1 (3) | 9.4 (3) |
| University | 58.6 (17) | 78.8 (26) | 75.0 (24) |
| Currently in paid job | | | |

**Table 1**  Continued

|  | % (N) | % (N) | % (N) |
| --- | --- | --- | --- |
| No | 20.7 (6) | 45.5 (15) | 25.0 (8) |
| Yes | 75.9 (22) | 54.5 (18) | 75.0 (24) |

HDL, high-density lipoprotein; LDL, low-density lipoprotein.

by waist circumference or per cent body fat. There were also no significant differences in change in fasting glucose or systolic and diastolic blood pressure among the three different fat interventions. In pairwise comparison, coconut oil significantly lowered C reactive protein compared to olive oil but there were no significant differences between coconut oil and butter for C reactive protein.

The results were somewhat surprising for a number of reasons. Coconut oil is predominantly (approximately 90%) saturated fat which is generally held to have an adverse effect on blood lipids by increasing blood LDL-C concentrations. However, the saturated fatty acid profiles of different dietary fats vary substantially; coconut oil is predominantly (around 48%) lauric acid (12:0) compared with butter (66% saturated fat) which is about 40% palmitic (16:0) and stearic (18:0) acids, leading to suggestions that coconut oil may not have the same health effects as other foods high in saturated fat.[9] Nevertheless, though reviews on coconut oil and cardiovascular disease risk factors have concluded that the evidence of an association between coconut oil consumption and blood lipids or cardiovascular risk was mostly poor quality,[9] trials have generally reported that coconut oil consumption raises LDL-C in comparison to polyunsaturated oil such as safflower oil, though not as much in comparison to butter.[10 11]

Based on three randomised crossover trials of good scientific quality, one trial reported butter increased LDL-C more than coconut oil which raised LDL-C more compared with safflower oil[10]; a second reported that coconut oil raised LDL-C more than beef fat which raised LDL-C more than safflower oil[23] and a third reported that coconut oil raised LDL-C more than palm oil which raised LDL-C more than olive oil.[24] The current study observed that butter raised LDL-C more than coconut oil but that coconut oil did not differ from olive oil. Two studies showed higher HDL-C with coconut oil compared with other fats whether beef fat, safflower oil or olive oil.[23 24] Thus far, the current results are consistent with previous studies indicating that butter raises LDL-C more than coconut oil and also that coconut oil also raises HDL-C. However, the present study is an exception in not finding any increase in LDL-C compared with an unsaturated oil, in this case, olive oil. In this trial, the difference of 0.33 mmol/L in LDL-C on butter compared with olive oil is consistent with previous studies.[25]

This is the largest trial reported to date on coconut oil and lipids apart from a recent study of 200 individuals with established coronary heart disease comparing coconut oil with sunflower oil over 2 years that reported no differences in blood lipids but virtually all the participants were on statin therapy[26] which makes findings difficult to interpret.

Direct comparisons between studies are problematic because of different oils used; we used extra virgin olive oil as a comparison group rather than a polyunsaturated oil such as safflower or sunflower oil, for feasibility reasons of likely participant compliance with the requirement for 50 g intake daily. The PREDIMED study reported no significant difference in change in LDL-C or TC but significant lowering of the cholesterol/HDL-C ratio in the Mediterranean diet supplemented with extra virgin olive oil compared with a low-fat diet.[2 13] A recent review reported that high phenolic olive oil does not modify the lipid profile compared with its low phenolic counterpart[14] though other studies have reported that extra virgin olive oil decreases LDL-C directly measured as concentrations of apoB-100 and the total number of LDL particles as assessed by NMR spectroscopy.[27 28] We therefore expected coconut oil would raise LDL-C compared with olive oil, but in the current study, we observed no evidence of an overall average increase in LDL-C in individuals allocated either to the coconut oil or olive oil intervention.

Lack of compliance with consuming the dietary fat would lead to no differences between groups and hence explain the lack of differences in LDL-C between coconut oil and olive oil groups. However, in this group of volunteers, reported compliance was high and did not differ between groups; in addition, those in the coconut oil group had significantly greater increases in HDL-C compared with those allocated to olive oil or butter, so lack of compliance is unlikely to be an explanation.

The predominant fatty acid in coconut oil, lauric acid (C12:0) as well as myristic acid (C14:0) are medium chain fatty acids that are rapidly absorbed, taken up by the liver and oxidised to increase energy expenditure which is a possible explanation for why coconut oil may have different effects compared with other saturated fats.[29] It is also possible that differences could be attributed to the use of extra virgin preparations of coconut oil rather than standard coconut oil; different methods of preparation such as the chilling method for virgin coconut oil compared with refined, bleached and deodorised coconut oil may influence phenolic compounds and antioxidant activity,[30] thus, processing of oils changes their composition, biological properties and consequent potential metabolic effects. The variations in possible health effects resulting from variations in processing of different fats is well documented in the

**Table 2**  Mean change in variables between baseline and follow-up after dietary interventions and pairwise comparisons between fats in 91 participants

| | Change from baseline | | | | Pairwise comparisons | | |
| | Coconut oil | Butter | Olive Oil | P value | Coconut oil vs olive oil | Butter vs Coconut oil | Butter vs olive oil |
| | n=28 | n=33 | n=30 | Comparison between groups | | | |
| | Mean (SD) | Mean (SD) | Mean (SD) | | Difference (95% CI) | Difference (95% CI) | Difference (95% CI) |
|---|---|---|---|---|---|---|---|
| LDL–cholesterol (mmol/L) | −0.09 (0.49) | 0.33 (0.48) | −0.06 (0.39) | <0.001 | −0.04 (−0.27 to 0.19) | 0.42 (0.19 to 0.65) | 0.38 (0.16 to 0.60) |
| Total cholesterol (mmol/L) | 0.22 (0.55) | 0.42 (0.59) | 0.03 (0.43) | 0.022 | 0.19 (−0.08 to 0.46) | 0.19 (−0.08 to 0.45) | 0.38 (0.11 to 0.64) |
| HDL–cholesterol (mmol/L) | 0.28 (0.29) | 0.09 (0.27) | 0.10 (0.15) | 0.009 | 0.16 (0.03 to 0.28) | −0.18 (−0.30 to −0.06) | −0.02 (−0.14 to 0.09) |
| Triglycerides (mmol/L) | 0.07 (0.58) | −0.001 (0.36) | −0.03 (0.27) | 0.65 | 0.10 (−0.12 to 0.32) | −0.08 (−0.29 to 0.13) | 0.02 (−0.19 to 0.23) |
| Cholesterol/HDL ratio | −0.26 (0.36) | 0.10 (0.41) | −0.13 (0.32) | <0.001 | −0.14 (−0.33 to 0.05) | 0.36 (0.18 to 0.54) | 0.22 (0.04 to 0.40) |
| Non HDL–cholesterol (mmol/L) | −0.06 (0.44) | 0.33 (0.51) | −0.07 (0.42) | 0.001 | 0.002 (−0.23 to 0.24) | 0.39 (0.16 to 0.62) | 0.39 (0.16 to 0.62) |
| Glucose (mmol/L) | −0.05 (0.49) | 0.02 (0.48) | −0.06 (0.49) | 0.68 | 0.01 (−0.23 to 0.25) | 0.08 (−0.15 to 0.32) | 0.09 (−0.14 to 0.33) |
| C reactive protein (mg/L) | −0.31 (1.09) | −0.04 (0.93) | 0.23 (1.40) | 0.11 | −0.58 (−1.12 to −0.04) | 0.29 (−0.24 to 0.82) | −0.29 (−0.80 to 0.23) |
| Weight (kg) | 0.27 (0.77) | 0.04 (1.00) | −0.04 (0.84) | 0.42 | 0.30 (−0.16 to 0.76) | −0.22 (−0.67 to 0.23) | 0.08 (−0.36 to 0.52) |
| Waist (cm) | 1.29 (3.31) | 0.26 (3.43) | 0.59 (3.25) | 0.52 | 0.71 (−1.00 to 2.42) | −0.95 (−2.63 to 0.72) | −0.24 (−1.89 to 1.41) |
| Body fat (%) | 0.24 (1.03) | 0.34 (1.31) | 0.13 (1.30) | 0.82 | 0.09 (−0.54 to 0.73) | 0.10 (−0.52 to 0.72) | 0.19 (−0.42 to 0.81) |
| Body mass index (kg/m²) | 0.09 (0.27) | 0.02 (0.35) | −0.01 (0.29) | 0.13 | 0.10 (−0.06 to 0.26) | −0.07 (−0.22 to 0.09) | 0.03 (−0.12 to 0.18) |
| Systolic blood pressure (mm Hg) | 0.18 (11.46) | −3.79 (11.11) | −3.67 (8.23) | 0.29 | 3.91 (−1.22 to 9.04) | −3.22 (−8.26 to 1.82) | 0.69 (−4.26 to 5.64) |
| Diastolic blood pressure (mm Hg) | −2.02 (5.71) | −1.33 (6.24) | −0.45 (8.48) | 0.81 | −0.73 (−3.88 to 2.42) | 0.99 (−2.08 to 4.05) | 0.26 (−2.78 to 3.30) |

HDL, high–density lipoprotein; LDL, low–density lipoprotein.

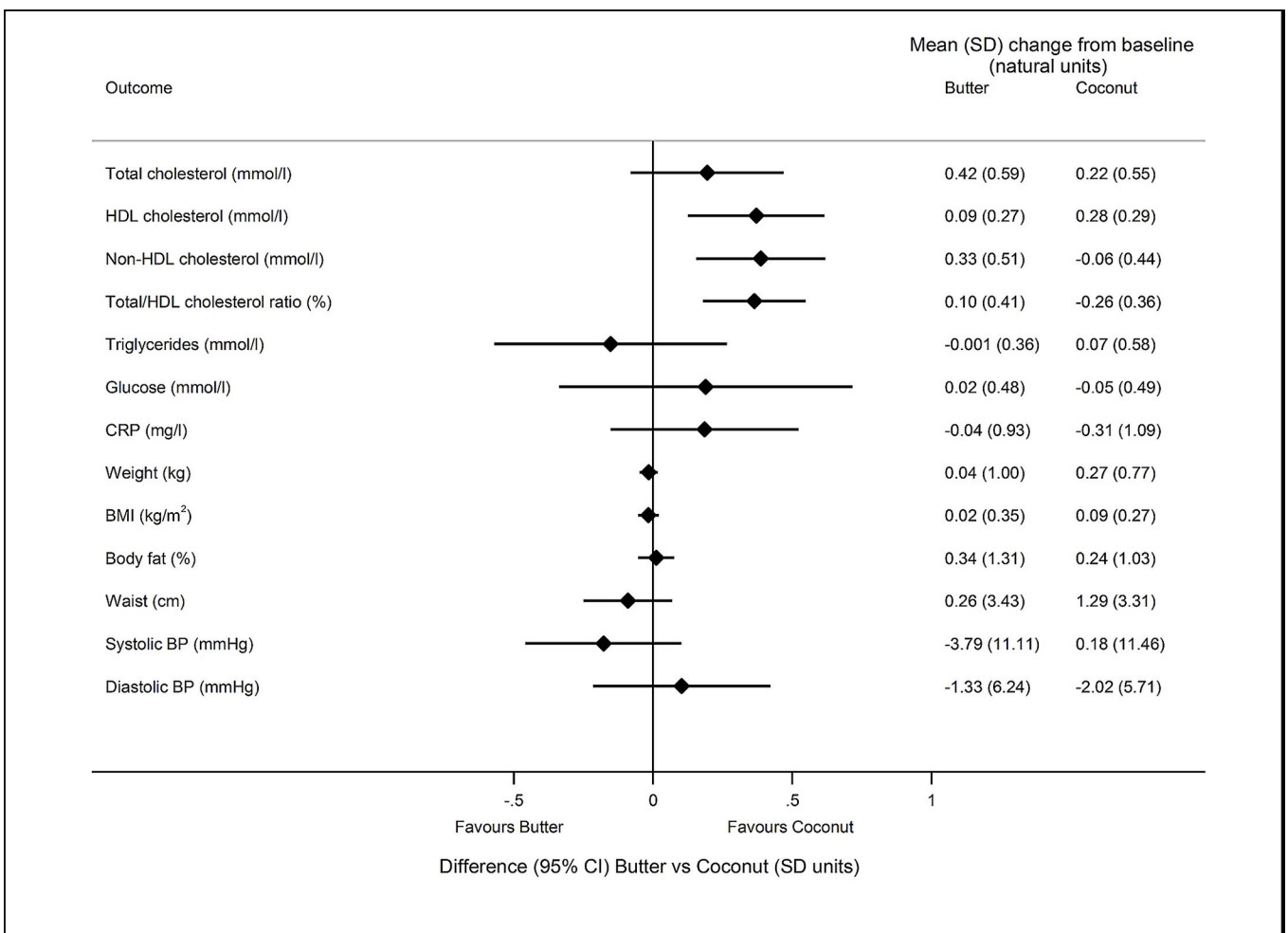

**Figure 2** Difference (95% CI) in the primary outcome (LDL cholesterol) between each pair of randomised groups, reported in units of baseline SD. Mean (SD) change from baseline is also presented for each group in mmol/L. COB study, intention-to-treat population, n=91. BMI, body mass index; BP, blood pressure; CRP, C reactive protein; HDL, high-density lipoprotein; LDL, low-density lipoprotein.

large literature on hydrogenation of polyunsaturated oils to make solid margarines which may increase harmful trans fats.[31]. In this context, it is notable that the major trial (PREDIMED) reporting reduction in cardiovascular risk with a Mediterranean diet used extra virgin olive oil,[2] while other studies which reported null findings with olive oil may not have always specified the product used.[14]

There was no evidence of difference between groups in mean weight, BMI, per cent body fat or central adiposity at the end of this trial; however, these were secondary endpoints for which the trial was not specifically powered. Nevertheless, the estimated 95% CI around mean weight differences at the end for the trial were not large. The participants were asked to consume 50 g of fat or oils daily. They could do this in the context of their usual diet by substituting for their usual fats or by consuming these as a supplement. In practice, most participants reported finding it difficult to substitute the different fats or oils for cooking in their usual diet and usually consumed these as a supplement. These fats if taken in addition to their usual diet would have been approximately 450 additional calories daily, which if consistently taken over 4 weeks

might be expected to be nearly 13 000 additional calories resulting in likely weight gain of 1–2 kg. This information was provided in the information sheet with the informed consent for participants. While it is possible that participants may have consciously changed behaviours to maintain body weight such as reducing their other dietary intake because of the additional fat or being more physically active, many participants reported that the high-fat diet resulted in feeling full and eating less.

It is also possible that even though this was a randomised trial, in an unblinded study, participants may have changed behaviours differentially in the different intervention groups resulting in differences in lipids or lack of differences in weight observed rather than being attributed to the dietary fat interventions. The majority of the participants reported no change in usual physical activity though slightly more participants in the coconut oil and butter groups reported increasing usual physical activity (14% and 15%, respectively) compared with 4% in the olive oil group. Nevertheless exclusion of all individuals reporting increased usual physical activity from the analyses did not change the findings. Dietary factors

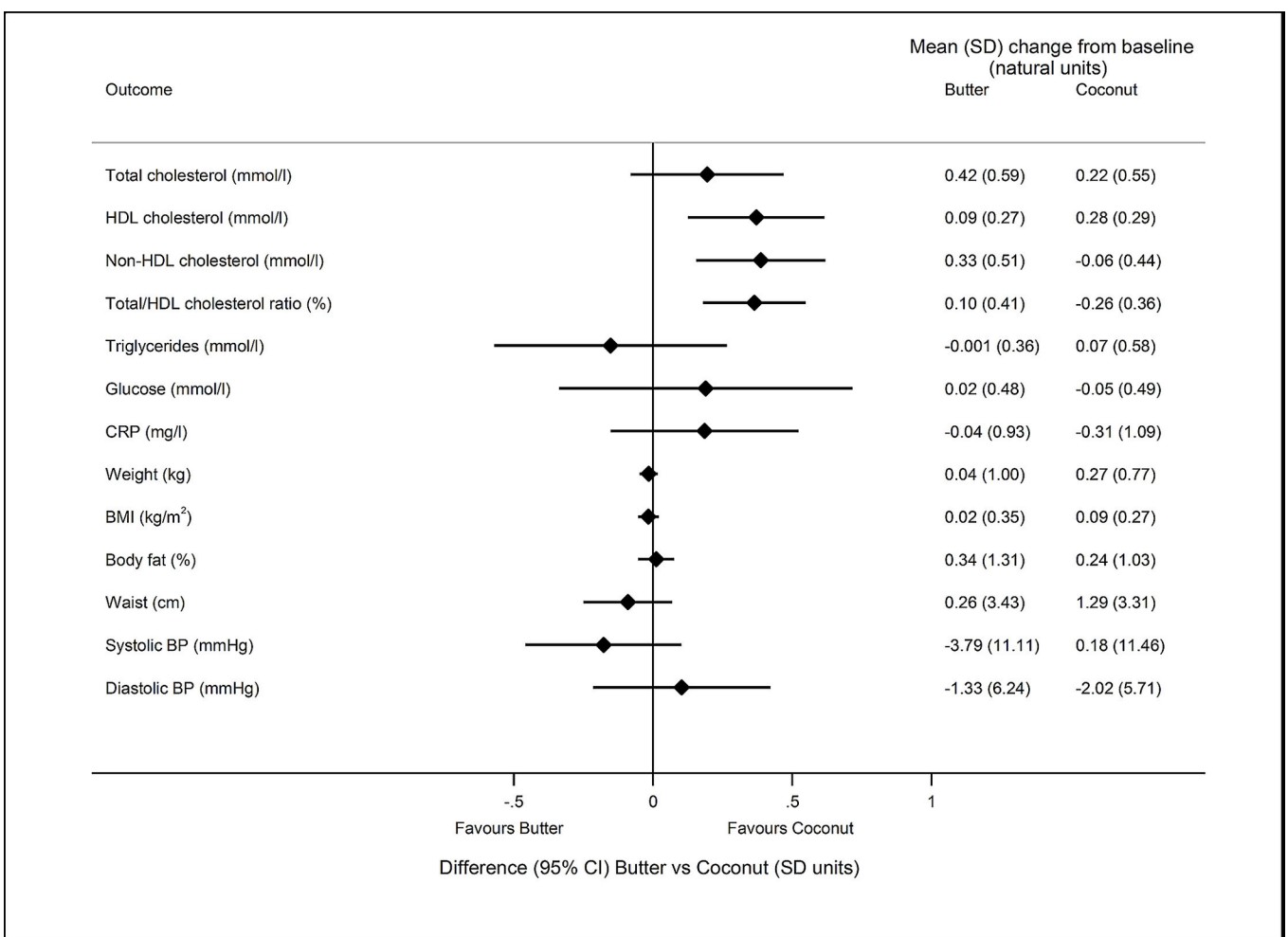

**Figure 3** Difference (95% CI) in secondary outcomes comparing butter vs coconut oil groups, reported in units of baseline SD. Mean (SD) change from baseline is also presented for each group in the natural units of the outcome. COB study, intention-to-treat population, n=91. For HDL cholesterol, sign of difference and 95% CI is the opposite of that reported in table 2, on the assumption that higher HDL is better, so the negative estimated difference (butter vs coconut) reported in table 2 is presented on the side of the graph which favours the coconut group. BMI, body mass index; BP, blood pressure; CRP, C reactive protein; HDL, high-density lipoprotein; LDL, low-density lipoprotein.

apart from fat most likely to influence HDL-C, total alcohol intake or change in alcohol intake did not differ significantly between intervention groups and in fact alcohol intake decreased slightly during the trial which would not explain any increases in HDL-C observed. There is therefore no evidence to suggest that differences in lipids or lack of differences in weight change were likely to be attributed to differential changes in behaviour.

The main strengths of this study are the randomised design with high completion rate (91/94 individuals returned to follow-up) and self-reported dietary compliance (nearly 90% participants with over 75% adherence) over 4 weeks. This is also larger than most trials reported with the exception of the trial in India in individuals with heart disease most of whom were taking statins[26]. The current trial by contrast was conducted in individuals in the general population.

This trial has limitations. It was a short-term trial of 4 weeks intervention, so we are unable to know what would have happened if the intervention had continued

for a longer period. Moreover, the current findings only apply to the intermediate metabolic (lipid) risk markers and cannot be extended to findings for clinical endpoints.

It was designed as a pragmatic trial in free living individuals rather than a controlled metabolic ward trial such that individuals were asked only to consume the 50 g of allocated fat or oil daily. As this was a 'real-world' study, we made no attempt to control other aspects of their usual diet in particular, total energy intake. For this reason, our results cannot be taken to reflect what would happen when the only change to a diet is the substitution of one fat with another (eg, replacing butter with coconut oil or replacing butter with olive oil). Individuals may have changed their behaviours in different ways to accommodate this additional fat, whether by modifying other aspects of their diet for instance, increasing foods such as bread and potatoes or salads to eat with the fats or consciously reducing other food intake or changing physical activity patterns to control energy balance. Nevertheless, this trial is more reflective of real-life situations.

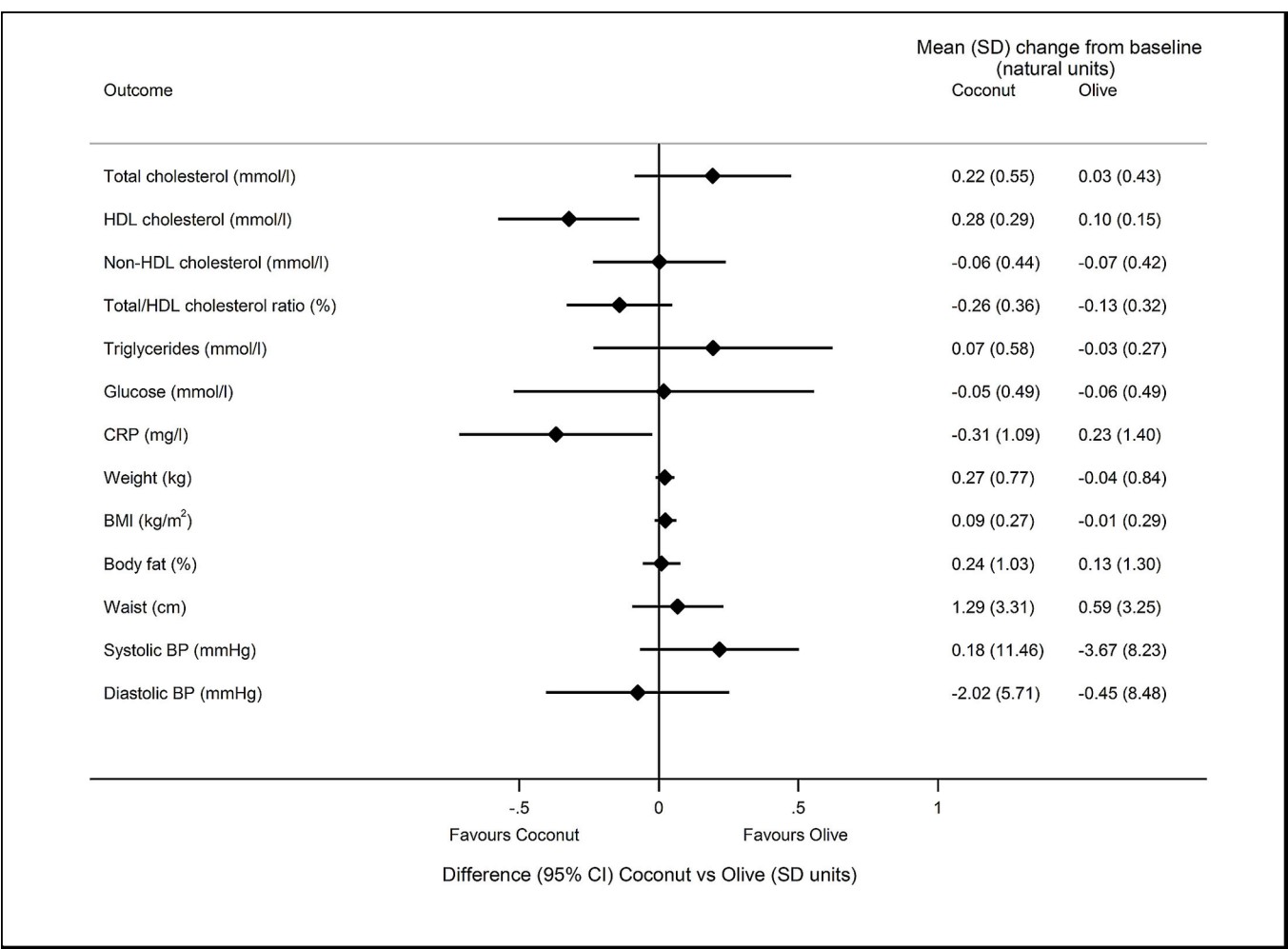

**Figure 4** Difference (95% CI) in secondary outcomes comparing coconut oil vs olive oil groups, reported in units of baseline SD. Mean (SD) change from baseline is also presented for each group in the natural units of the outcome. COB study, intention-to-treat population, n=91. For HDL-cholesterol, sign of difference and 95% CI is the opposite of that reported in table 2, on the assumption that higher HDL is better, so the positive estimated difference (coconut vs olive) reported in table 2 is presented on the side of the graph which favours the coconut group. BMI, body mass index; BP, blood pressure; CRP, C reactive protein; HDL, high-density lipoprotein; LDL, low-density lipoprotein.

While self-reported compliance was high, this was subjective and we did not measure the blood fatty acid profile in participants following the intervention for an objective biomarker of compliance. Nevertheless, we did observe differential changes in blood lipids during the intervention.

The generalisability of the findings to the wider population is also unclear. The volunteers were clearly highly selected to be willing to participate in such a study and also likely to be healthier than the general population, as for ethical reasons we excluded those with known prevalent cardiovascular disease, cancer or diabetes and also those on any lipid lowering medication or other contraindications to a high-fat diet. Nevertheless, it is unlikely that the effect of these dietary fats in this group of individuals recruited from the general population would be biologically different from the general population.

**Implications**

We focused on LDL-C for the primary endpoint as the causal relationship between LDL-C concentrations and coronary heart disease risk is well established, with about a 15% increase in coronary heart disease risk per 1 mmol/L increase in LDL-C concentrations and reduction of LDL-C cholesterol lowers coronary heart disease risk.[32] Increase in LDL-C concentrations has been the main mechanism through which dietary saturated fat is believed to increase heart disease risk, though other pathways have been postulated. However, it is notable that some Mediterranean diet interventions such as the Lyon heart study (alpha linolenic acid)[33] or PREDIMED (extra virgin olive oil)[2] which have been reported to reduce cardiovascular risk in secondary and primary prevention may have effects through other pathways such as inflammation or endothelial function.[34 35] Whatever the mechanisms, the evidence from prospective studies is consistent and strong that substitution of saturated fats by unsaturated fats is beneficial for cardiovascular risk.[36]

The results of this study indicate that two different dietary fats (coconut oil and butter), which are predominantly saturated fats, appear to have different effects

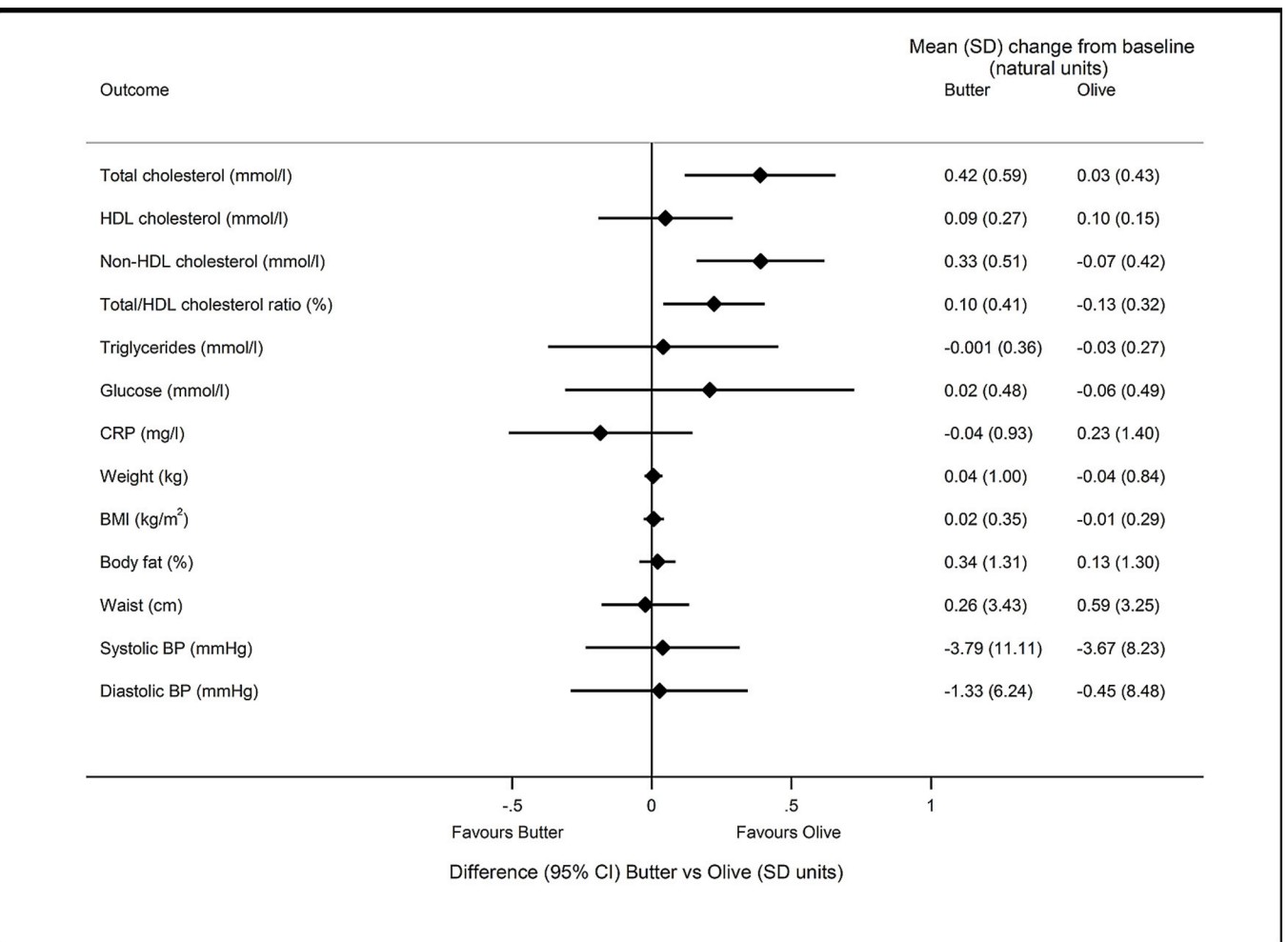

**Figure 5** Difference (95% CI) in secondary outcomes comparing butter vs olive oil groups, reported in units of baseline SD. Mean(SD) change from baseline is also presented for each group in the natural units of the outcome. COB study, intention-to-treat population, n=91. For HDL-cholesterol, sign of difference and 95% CI is the opposite of that reported in table 2, on the assumption that higher HDL is better, so the negative estimated difference (butter vs olive) reported in table 2 is presented on the side of the graph which favours the olive group. BMI, body mass index; BP, blood pressure; CRP, C reactive protein; HDL, high-density lipoprotein; LDL, low-density lipoprotein.

on blood lipids compared with olive oil, a predominantly monounsaturated fat. The effects of different dietary fats on lipid profiles, metabolic markers and health outcomes may vary not just according to the general classification of their main component fatty acids as saturated or unsaturated but possibly according to different profiles in individual fatty acids, processing methods as well as the foods in which they are consumed or dietary patterns. There is increasing evidence that associations of saturated fatty acids with health outcomes may vary according to whether they are odd or even chain saturated fatty acids or their chain length.[37–39] Indeed, while overall the evidence indicates the substitution of dietary saturated fats with polyunsaturated fats is beneficial for coronary heart disease risk[40] heterogeneity in findings from observational studies and trials may reflect different dietary sources of fats.[4 41] As the Joint FAO/WHO 2008 Expert Consultation on Fats and Fatty Acids in Human Nutrition comments:

*'There are inherent limitations with the convention of grouping fatty acids based only on number of double bonds…major groups of fatty acids are associated with different health effects…individual fatty acids within each broad classification may have unique biological properties or effects…Intakes of individual fatty acids differ across world depending on predominant food sources of total fats and oils.'* The associations with health endpoints may well vary depending on the food sources.

In this trial, extra virgin coconut oil was similar to olive oil and did not raise LDL-C in comparison with butter. The current short-term trial on an intermediate cardiovascular disease risk factor, LDL-C, does not provide evidence to modify existing prudent recommendations to reduce saturated fat in the diet as emphasised in most consensus recommendations[8 12] and dietary guidelines should be based on a range of criteria.[42] However, the findings highlight the need for further elucidation of the more nuanced relationships between different dietary fats and health. There is increasing evidence that to understand

**Table 3** Baseline and follow-up dietary intake by allocation to coconut oil, butter or olive oil* estimated using 24-hour DietWebQ

| DietWebQ intake/day | Coconut oil | Butter | Olive oil |
|---|---|---|---|
| Baseline prior to start of intervention | n=27 | n=33 | n=32 |
| Energy (MJ/day) | 9.0 (3.7) | 8.2 (2.2) | 9.5 (3.5) |
| Total fat (g/day) | 94 (47) | 81 (26) | 98 (50) |
| Protein (g/day) | 74 (29) | 75 (19) | 87 (34) |
| Carbohydrate (g/day) | 238 (95) | 215 (75) | 243 (95) |
| Alcohol (g/day) | 16 (22) | 17 (23) | 18 (22) |
| At 4 weeks of intervention | n=24 | n=32 | n=27 |
| Energy (MJ/day) | 9.6 (3.2) | 8.6 (2.4) | 9.6 (3.1) |
| Total fat (g/day) | 127 (47) | 94 (37) | 138 (38) |
| Protein (g/day) | 71 (25) | 77 (29) | 78 (31) |
| Carbohydrate (g/day) | 215 (84) | 214 (64) | 197 (101) |
| Alcohol (g/day) | 9 (15) | 13 (15) | 8 (18) |
| Change from baseline | n=24 | n=32 | n=27 |
| Energy (MJ/day) | 0.3 (2.9) | 0.5 (2.0) | −0.4 (2.8) |
| Total fat (g/day) | 29 (43) | 14 (36) | 28 (40) |
| Protein (g/day) | −7 (33) | 3 (30) | −12 (26) |
| Carbohydrate (g/day) | −31 (74) | 4 (69) | −55 (81) |
| Alcohol (g/day) | −8 (22) | −5 (23) | −11 (27) |

*Numbers do not total 94 as not all participants completed the baseline and follow-up DietWebQ.

the relationship between diet and health, we need to go beyond simplistic associations between individual nutrients and health outcomes and examine foods and dietary patterns as a whole. In particular, present day diets with high intakes of processed foods now incorporate many fats and oils such as soya bean oil, palm oil and coconut oil which have not been previously widely used in Western societies and not well studied. The relationships between different dietary fats, particularly some of the now more commonly used fats, and health endpoints such as cardiovascular disease events need to be better established.

**Acknowledgements** This study was conducted in collaboration with the British Broadcasting Corporation (BBC) which provided support for the recruitment of participants, running of the community assessment clinic and biochemistry measurements for lipids. Other costs were supported by the University of Cambridge through a National Institute of Health Research Senior Investigator Award to K-TK. NGF acknowledges core MRC Epidemiology Support (MC UU 12015/5). We thank Keith Burling and Peter Barker from the Core Biochemical Assay Laboratory, CBAL in Cambridge for the laboratory assays, Shrikant Bangdiwala, University of North Carolina for conducting the computer generated random allocation of participants to the interventions, Timothy Key and colleagues at Oxford University for the use of the DietWebQ, and Nichola Dalzell and Shabina Hayat, Department of Public Health and Primary Care, and Eirini Trichia, Richard Powell and Meriel Smith, MRC Epidemiology Unit, University of Cambridge for logistical support. We thank the Cambridge Yoga Centre which hosted the assessment sessions for participants in June and July 2017. Most of all, we thank the participants from the general community who generously volunteered to take part in this trial; this study would not have been possible without their efforts and we are most grateful to them.

**Contributors** K-TK had full access to all of the data in the study and takes responsibility for the integrity of the data and the accuracy of the data analysis. Study concept and design: KT-K, NGF, LF. Acquisition of data: KT-K, NGF, LF, IA, RL, ML. Analysis and interpretation of the data: KT-K, NGF, LF. Drafting of the manuscript: KT-K. Critical revision of the manuscript for important intellectual content: NGF, SJS, IA, LF, RL, ML. Obtaining funding: KT-K, NGF, LF. Administrative, technical or material support: KT-K, NGF, LF, IA, RL, SJS, ML.

**Funding** This work was supported by the British Broadcasting Corporation, a National Institute of Health Research Senior Investigator Award to K-TK and core MRC Epidemiology support (MC UU 12015/5).

**Disclaimer** The lead author and guarantor K-TK affirms that the manuscript is an honest, accurate and transparent account of the study being reported; that no important aspects of the study have been omitted and that any discrepancies from the study as planned have been explained.

**Competing interests** None declared.

**Patient consent** Obtained.

**Ethics approval** Ethics approval was given for the study by the University of Cambridge Human Biology Research Ethics committee HBREC 2017.05.

**Provenance and peer review** Not commissioned; externally peer reviewed.

**Data sharing statement** Data are available. Please contact corresponding author.

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
