## [Reviewer comments · BMJ Open]

This paper was submitted to a another journal from BMJ but declined for publication following peer review. The authors addressed the reviewers' comments and submitted the revised paper to BMJ Open. The paper was subsequently accepted for publication at BMJ Open.

This paper received five reviews from its previous journal but only three reviewers gave the permission to publish their review.

ARTICLE DETAILS

TITLE (PROVISIONAL)	Randomized trial of coconut oil, olive oil or butter on blood lipids and other cardiovascular risk factors in healthy men and women
AUTHORS	Khaw, Kay-Tee; Sharp, Stephen; Finikarides, Leila; Afzal, Islam; Lentjes, Marleen; Luben, Robert; Forouhi, Nita

VERSION 1 – REVIEW

REVIEWER	Ros, Emilio Institut d'Investigacions Biomèdiques August Pi i Sunyer (IDIBAPS) I have included my name in my comments for the author
REVIEW RETURNED	15-Aug-2017

GENERAL COMMENTS	This paper reports the results of a parallel group, randomized nutrition intervention trial in apparently healthy volunteers from the community aged 50 to 75 y testing the intake of 50 g daily of extra virgin coconut oil (EVCO), extra virgin olive oil (EVOO) or unsalted butter added to their usual diets for 4 weeks. Main outcome was changes in LDL cholesterol (LDL-C) and other lipid fractions and cardiovascular risk factors were secondary outcomes. This trial stems from public interest in the health effects of EVCO as voiced by a BBC audience. Results show that, similar to EVOO, EVCO consumption does not increase LDL-C or non-HDL-C, while butter predictably does. EVCO also appears to rise HDL-C to a greater extent than either butter or EVOO. This is another coconut oil trial comparing lipid effects of this saturated fat rich oil with those of other oils or fats and have had inconsistent results, as summarized in the 2016 revision of Eyres et al. (ref. 9 of present paper). While present results seem to favour EVCO for not rising LDL-C and increasing HDL-C and thus differ from those of the only three scientifically sound trials included in that revision (see comments below), they cannot be taken as sound proof that consumption of EVCO is beneficial for cardiovascular health because of some pitfalls of the study. Besides, the paper is difficult to read due to cumbersome syntax, the background on prior evidence regarding the oils tested is insufficient, and some assertions regarding EVOO are incorrect. Authors should pay attention to the following points:
---

Major

1. In a nutrition intervention trial it is critical to document any dietary changes from baseline beyond those attributable to the tested foods or nutrients. Here only a baseline 24 h food record was administered.

Besides the fact that a 24 h food record is insufficient to document usual diet (a standard food frequency questionnaire or several 24 h food records are in order), no follow-up questionnaire was obtained. Because widely different supplemental fats were given without advice on food substitution, compensation by avoiding other foods could differ between treatment arms and this would have an impact on nutrient intake affecting the investigated endpoints beyond the tested fats. This should be acknowledged as a limitation to the study.

2. No biomarkers of fat intake were measured to objectively ascertain compliance. For instance, blood fatty acids for changes in lauric acid (EVCO), odd chain fatty acids (butter) or oleic acid (EVOO), while imperfect, could have provided some insight on compliance. This is another limitation.

3. A Table disclosing the fatty acid composition of the tested fats would be useful to the non-expert reader.

4. It is stated that the trial took place between June and July 2017. How can a single nutritionists deal with 90+ participants evaluated at baseline and after 4 weeks in a single month? This incredibly short time defeats comprehension for anybody used to perform nutrition intervention studies.

5. Concerning prior intervention trials comparing coconut oil to other fats for lipid effects, the Eyres et al. review is right in concluding that the evidence is of very poor quality. However, in their review of 8 studies published up to end of 2013 (no good quality studies have been published since), only 3 randomized crossover trials can be considered of good scientific quality (3 of the reviewed trials were sequential – totally unscientific! -, one had only 9 participants, and another used no comparator oils or fats). The summary of LDL-C changes from the 3 sound trials is: butter > coconut oil > safflower oil in one trial; coconut oil > beef fat > safflower oil in another; and coconut oil > palm oil > olive oil in the third. The results of the present study show LDL-C changes: butter > coconut oil = olive oil. HDL-C was higher with coconut oil in two studies and not different compared with other fats in another study; HDL-C was also higher with coconut oil in the present study. Considering all these studies, two show that butter raises LDL-C more than coconut oil and two show that coconut oil raises LDL-C more than other oils or fats rich in saturated fats, and three show that coconut oil raises LDL-C more than cis unsaturated oils, the present study being an exception. To summarise, the evidence thus far is that coconut oil raises LDL-C, although not as much as butter does, while it also increases HDL-C. The Discussion would benefit from a consideration of these overall results.

6. The saturated fatty acids of coconut oil, primarily lauric acid (C12:0) and myristic acid (C14:0), are medium chain fatty acids that are rapidly absorbed, taken up by the liver and oxidized to increase energy expenditure (DeLany JP, et al. Am J Clin Nutr 2000;72:905–11).

This can be stated when purporting to describe differential effects of coconut oil and other saturated oils or fats.

7. There are incorrect statements on the lipid effects of EVOO, newer evidence not cited, and findings relating to the PREDIMED study that are also incorrectly cited (page 5, lines 8-13; page 12, lines 54-55; page 14, lines 43-45; and page 15, lines 27-28). Contrary to what is stated in the manuscript, the report of ref. 13 indeed shows a reduction of LDL-C and an HDL-C increase by the Mediterranean diet supplemented with EVOO compared to the control diet. While the meta-analysis of ref. 14 concludes that high-phenolic olive oil does not modify the lipid profile compared to its low-phenolic counterpart, only studies published up to July 2013 are dealt with. Recent studies show that EVOO decreased LDL concentrations directly measured as concentrations of apo B-100 and the total number of LDL particles as assessed by NMR spectroscopy (Hernaiz A, et al. *J Nutr.* 2015;145:1692-7.) and that polyphenol-rich olive oil enhances HDL function compared with phenol-poor olive oil (Hernández A, et al. *Olive oil polyphenols enhance high-density lipoprotein function in humans: a randomized controlled trial. Arterioscler Thromb Vasc Biol.* 2014;34:2115–19). A prior intervention trial (EUROLIVE) also showed that olive oil polyphenols dose-dependently increased HDL-C levels (Covas MI, et al. *Ann Intern Med.* 2006;145:333–41).

8. In the statistical approach (page 8) there is no description of statistical tests used for between-group comparisons (i.e., ANOVA, as shown in Table 2). Adjustment for multiple testing should be considered.

9. Results, bottom of page 10 to top of page 11. What do authors mean by feeling better or worse consuming the different fats? What questions were asked or what tests were administered?

10. Discussion, page 13, lines 17-24. The evidence is incontrovertible that processing of oils profoundly changes their composition, mainly by practically eliminating phenolic compounds. Authors should use stronger affirmative wording here.

11. Discussion, page 16, top. The evidence from prospective studies is consistent and strong that substitution of saturated fats for polyunsaturated (and monounsaturated) fats is beneficial for cardiovascular risk. Besides ref. 26, please cite and comment Li Y et al. *J Am Coll Cardiol* 2015;66:1538–48. The Sydney Diet Heart trial (ref. 27) is not a good reference to suggest that polyunsaturated fats may be harmful because it was carried out in the 1970s and the poly fat given in that trial was part of a margarine that was presumably rich in deleterious trans fatty acids at a time when they were little known. Thus, any benefit from poly fatty acids was counteracted by concomitant trans fatty acids.

Minor

11. Page 4, line 19. Verb missing.

13. Page 4, line 42. ... predominant in butter (add palm oil and animal fat).

14. Page 11. Line 13. ... "versus butter" should be omitted.

15. Page 13, line 24. The sentence ending in ref. (21) should be omitted. Trans fat production by hydrogenation of margarines bears no relationship to processing of edible oils.

	16 Id, line 31. Cite ref. 2 here. 17. Page 14, line 24. Second colon misplaced. 18. Id, lines 26-31. That individuals may have changed their behaviours in different ways to accommodate additional fats is a real fact of daily (eating) life, not a limitation to the study. 19. Id, lines 45-49. Which fat at a daily 50 g dose can raise LDL-C levels as shown in metabolic ward studies? Butter? 20. Id, sentence at bottom. Authors should discuss the concept of energy compensation via food displacement, well demonstrated in trials of other high fat foods such as nuts. 21. Bottom page 14 to top page 15. It is unclear for which ethical reasons one must excluded patients with known prevalent cardiovascular disease, cancer or diabetes when performing a short-term nutrition intervention trial like this one. Also there is no clear reason to exclude those on lipid lowering medication. Neither prior systemic disease or lipid lowering medication are formal contraindications to a high fat diet, even less if only followed for just 4 weeks. 22. Page 15, line 20. A reference is needed for the sentence ending "clearly other pathways have been postulated". Again, please provide an appropriate reference in next par., lines 31-32. 23. Id. Line 22. Colon misplaced.
--	---

REVIEWER	Sun, Qi Harvard T.H. Chan School of Public Health I have included my name in my comments for the author
REVIEW RETURNED	16-Aug-2017

GENERAL COMMENTS	This is a randomized clinical trial conducted by a team of established investigators led by Dr. Khaw at Cambridge University. The trial compared the effects of three oils/fats, i.e., extra virgin coconut oil, butter, or extra virgin olive oil, on some cardiometabolic risk factors or markers. The main findings are that butter significantly increased LDL levels in comparison with the other two types of oils, whereas coconut oil significantly increased HDL levels in comparison with the other oils/fats. Overall, this is a rigorously-conducted clinical trial that addressed a critical research question. I have some thoughts for this excellent work, aiming to clarify certain issues. Major comments:  1. In the description of eligibility criteria for study participants, were they taking lipid-lowering medications or not? Reading the abstract, my impression is that all participants did not take such medications, whereas when reading the method section, I have the impression that they actually did. 2. The duration of fasting is known to influence blood lipid levels. The participants were required to fast for at least 4 hours. Have the authors considered stratifying the analysis by duration of fasting, say 4-8 hours vs ≥ 8 hours? 3. It is important to understand whether samples from different intervention groups were assayed in a randomized and balanced manner across batches. 4. The current trial is an open-label study. The participants knew what they were given. Is the self-reported compliance validated using biomarkers or other measures? 5. Is the 0.16 mmol/L difference in HDL levels between coconut oil and olive oil groups after the trial the difference between delta of HDL in each group or the difference of mean HDL levels after the intervention? It's unclear in the manuscript.
---

	6. It is interesting to observe that at study baseline participants who were randomized to coconut oil groups already had higher HDL levels (+0.2 mmol/L) than the olive oil group. Likewise, coconut oil group had lower LDL levels (-0.2 mmol/L) than olive oil group at baseline. It seems that the randomization did not generate balanced distribution of baseline blood lipid levels between the three intervention groups. 7. One limitation that deserves some attention. As total energy and dietary carbohydrate intake may also influence blood lipid levels, to test independent effects of different fats, intake of carbohydrates (% of energy) should be maintained in an isocaloric setting between groups. The current trial did not seek to maintain a comparable carbohydrate intake level, nor a constant calorie intake. Such a limitation deserves some further discussions.
--	--

REVIEWER	Michel de Lorgeril Grenoble School of Medicine I have included my name in my comments for the author
REVIEW RETURNED	This is an interesting but (too) naive article. To improve it, it would be important to clearly describe the full fatty acid composition of each tested fat, to measure the plasma and red cell fatty acid composition of each subject before and after intervention and then study potential association between each class (or each fatty acid) of fatty acid and the various endpoint

VERSION 1 – AUTHOR RESPONSE

Reviewer: 2

Comments:

This paper reports the results of a parallel group, randomized nutrition intervention trial in apparently healthy volunteers from the community aged 50 to 75 y testing the intake of 50 g daily of extra virgin coconut oil (EVCO), extra virgin olive oil (EVOO) or unsalted butter added to their usual diets for 4 weeks. Main outcome was changes in LDL cholesterol (LDL-C) and other lipid fractions and cardiovascular risk factors were secondary outcomes. This trial stems from public interest in the health effects of EVCO as voiced by a BBC audience. Results show that, similar to EVOO, EVCO consumption does not increase LDL-C or non-HDL-C, while butter predictably does. EVCO also appears to rise HDL-C to a greater extent than either butter or EVOO. This is another coconut oil trial comparing lipid effects of this saturated fat rich oil with those of other oils or fats and have had inconsistent results, as summarized in the 2016 revision of Eyres et al. (ref. 9 of present paper). While present results seem to favour EVCO for not rising LDL-C and increasing HDL-C and thus differ from those of the only three scientifically sound trials included in that revision (see comments below), they cannot be taken as sound proof that consumption of EVCO is beneficial for cardiovascular health because of some pitfalls of the study. Besides, the paper is difficult to read due to cumbersome syntax, the background on prior evidence regarding the oils tested is insufficient, and some assertions regarding EVOO are incorrect. Authors should pay attention to the following points:

Major

1. In a nutrition intervention trial it is critical to document any dietary changes from baseline beyond those attributable to the tested foods or nutrients. Here only a baseline 24 h food record was administered.

Besides the fact that a 24 h food record is insufficient to document usual diet (a standard food frequency questionnaire or several 24 h food records are in order), no follow-up questionnaire was obtained.

Because widely different supplemental fats were given without advice on food substitution, compensation by avoiding other foods could differ between treatment arms and this would have an impact on nutrient intake affecting the investigated endpoints beyond the tested fats. This should be acknowledged as a limitation to the study.

Response: We did collect dietary information both at baseline and at the end of follow-up, and have now shown these results in Table 3, and discussed the results. We have indeed discussed the limitations mentioned above (please see discussion section, page 16-17, lines***).

2. No biomarkers of fat intake were measured to objectively ascertain compliance. For instance, blood fatty acids for changes in lauric acid (EVCO), odd chain fatty acids (butter) or oleic acid (EVOO), while imperfect, could have provided some insight on compliance. This is another limitation.

Response: While we are planning to measure blood fatty acids in the participants, this has substantial cost and will take some time. We have discussed this limitation in the discussion. No other trials reported have assessed blood fatty acids.

3. A Table disclosing the fatty acid composition of the tested fats would be useful to the non-expert reader.

Response: We have now included a supplemental table on the fatty acid composition of the tested fats and have also included this information in the results. This confirms that Coconut oil was 94 % saturated fatty acids, of which the main components were lauric acid C12:0 (48%) and myristic acid C14:0 (19%), palmitic acid C16:0 (9%) and caprylic acid C8:0 (9%); and 5% mono unsaturated fat, mainly oleic acid C18:1n9 (5%). Butter was 66% saturated fatty acids, of which the main components were palmitic acid C16:0 (28%), stearic acid C18:0 (12%), myristic acid C14:0 (11%); 26% monounsaturated fat, mainly oleic acid C18:1n9 (22%); and 3% polyunsaturated fat, linoleic acid C18:2n6 (2%) and alpha-linolenic acid (1%). Olive oil was 19% saturated fatty acids, mainly palmitic acid C16:0, 15% with stearic acid C18:0 (3%); 68% monounsaturates with the main component being oleic acid C18:1n9 (64%); and 13% polyunsaturates Linoleic acid C18:2n6 (12%). These profiles are very similar to those reported from other studies.

4. It is stated that the trial took place between June and July 2017. How can a single nutritionists deal with 90+ participants evaluated at baseline and after 4 weeks in a single month? This incredibly short time defeats comprehension for anybody used to perform nutrition intervention studies.

Response: We are not sure whether the reviewer is referring to implausibility of the dietary evaluations by a single nutritionist or to the assessment of participants. The nutritionist led the analysis of the dietary data. we stated in the methods that the participants completed an online dietary assessment, the DietWebQ which has automated nutrient and food estimation so results are available at once. This is the method used in UK Biobank of half a million participants and other large epidemiologic studies. We referenced this method in the manuscript.

If the reviewer in this statement, considers that it is implausible that the baseline and follow up assessments could be completed during June and July 2017, we should clarify that the assessments were not conducted by the nutritionist but by other co authors on the manuscript as indicated in the author contributions.. Indeed we completed this intervention trial at great speed and efficiency. We had six days of test-site visits for the participants at baseline (over the precise work-day (non-weekend) dates of June 22nd till June 29th, 2017), and a further six days of test-site visits at follow-up (over the precise dates of July 20th to July 27th, 2017).

For baseline, with 94 participants attending, this meant an average of between 15-16 participants per day, across two researchers (thus each researcher (KTK and NGF) seeing upto 8 participants each day). This was perfectly feasible given that we invited participants in groups, over half hour slots between 8 am till 12.30 pm every day of testing, and the clinical testing included the clinic measures we described in the manuscript (anthropometry, blood pressure). A designated phlebotomist (IA) took the blood tests for each participant, and a BBC researcher (LF) administered the questionnaires and after the clinical testing and phlebotomy, she also handed over the intervention oil or fat with instructions. The volunteers were able to contact the BBC researcher (LF) with any questions or clarifications in the four-week intervention period, and any queries that needed further input were transferred to the lead researchers KTK and NGF. We had extremely positive feedback from the volunteers that the study was done very efficiently over 4 weeks.

We would like to add that for us (KTK, NGF) as researchers this was a refreshing experience ourselves – to have been able to see a study through from ethical approval in April 2017 to baseline data collection in June and follow-up data collection in July). Our lab that measured blood samples was also extremely efficient and completed all measurements within 4 days of the end of the data collection period.

5. Concerning prior intervention trials comparing coconut oil to other fats for lipid effects, the Eyres et al. review is right in concluding that the evidence is of very poor quality. However, in their review of 8 studies published up to end of 2013 (no good quality studies have been published since), only 3 randomized crossover trials can be considered of good scientific quality (3 of the reviewed trials were sequential – totally unscientific! -, one had only 9 participants, and another used no comparator oils or fats). The summary of LDL-C changes from the 3 sound trials is: butter > coconut oil > safflower oil in one trial; coconut oil > beef fat > safflower oil in another; and coconut oil > palm oil > olive oil in the third. The results of the present study show LDL-C changes: butter > coconut oil = olive oil. HDL-C was higher with coconut oil in two studies and not different compared with other fats in another study; HDL-C was also higher with coconut oil in the present study. Considering all these studies, two show that butter raises LDL-C more than coconut oil and two show that coconut oil raises LDL-C more than other oils or fats rich in saturated fats, and three show that coconut oil raises LDL-C more than cis unsaturated oils, the present study being an exception. To summarise, the evidence thus far is that coconut oil raises LDL-C, although not as much as butter does, while it also increases HDL-C. The Discussion would benefit from a consideration of these overall results.

Response: we have now included more detail on these previous studies in the discussion.

6. The saturated fatty acids of coconut oil, primarily lauric acid (C12:0) and myristic acid (C14:0), are medium chain fatty acids that are rapidly absorbed, taken up by the liver and oxidized to increase energy expenditure (DeLany JP, et al. *Am J Clin Nutr* 2000;72:905–11). This can be stated when purporting to describe differential effects of coconut oil and other saturated oils or fats.

Response: we have now included this in the discussion

7. There are incorrect statements on the lipid effects of EVOO, newer evidence not cited, and findings relating to the PREDIMED study that are also incorrectly cited (page 5, lines 8-13; page 12, lines 54-55; page 14, lines 43-45; and page 15, lines 27-28). Contrary to what is stated in the manuscript, the report of ref. 13 indeed shows a reduction of LDL-C and an HDL-C increase by the Mediterranean diet supplemented with EVOO compared to the control diet. While the meta-analysis of ref.

14 concludes that high-phenolic olive oil does not modify the lipid profile compared to its low-phenolic counterpart, only studies published up to July 2013 are dealt with.

Recent studies show that EVOO decreased LDL concentrations directly measured as concentrations of apo B-100 and the total number of LDL particles as assessed by NMR spectroscopy (Hernaez A, et al. *J Nutr.* 2015;145:1692-7.) and that polyphenol-rich olive oil enhances HDL function compared with phenol-poor olive oil (Hernández A, et al. Olive oil polyphenols enhance high-density lipoprotein function in humans: a randomized controlled trial. *Arterioscler Thromb Vasc Biol.* 2014;34:2115–19). A prior intervention trial (EUROLIVE) also showed that olive oil polyphenols dose-dependently increased HDL-C levels (Covas MI, et al. *Ann Intern Med.* 2006;145:333–41).

Response: we have addressed these points in the discussion and included the references. Reference 13 (Estruch et al *Ann Int Med* 2006) while it reported that LDL-C concentrations in the olive oil group after intervention were significantly lower compared to baseline, in the direct comparison of the olive oil group with the low fat control group, only HDL-C was significantly increased but LDL-C was not significantly lower.

8. In the statistical approach (page 8) there is no description of statistical tests used for between-group comparisons (i.e., ANOVA, as shown in Table 2). Adjustment for multiple testing should be considered.

Response: we have now included more detail on the statistical methods. We considered adjustment for multiple testing, but since this is an exploratory rather than confirmatory trial, we decided to follow the advice of Bender et al (*Journal of Clinical Epidemiology* 54 (2001) 343–349) that adjustments are not strictly necessary. The numerical p-values reported in Table 2 show that even applying a conservative adjustment would not affect the overall trial conclusions.

9. Results, bottom of page 10 to top of page 11. What do authors mean by feeling better or worse consuming the different fats? What questions were asked or what tests were administered?

Response: we have put more detail about these questions in the methods.

10. Discussion, page 13, lines 17-24. The evidence is incontrovertible that processing of oils profoundly changes their composition, mainly by practically eliminating phenolic compounds. Authors should use stronger affirmative wording here.

Response: we have used stronger wording here as suggested.

11. Discussion, page 16, top. The evidence from prospective studies is consistent and strong that substitution of saturated fats for polyunsaturated (and monounsaturated) fats is beneficial for cardiovascular risk. Besides ref. 26, please cite and comment Li Y et al. *J Am Coll Cardiol* 2015;66:1538–48. The Sydney Diet Heart trial (ref. 27) is not a good reference to suggest that polyunsaturated fats may be harmful because it was carried out in the 1970s and the poly fat given in that trial was part of a margarine that was presumably rich in deleterious trans fatty acids at a time when they were little known. Thus, any benefit from poly fatty acids was counteracted by concomitant trans fatty acids.

Response: we have included this in the discussion

Minor

11. Page 4, line 19. Verb missing.

13. Page 4, line 42. ... predominant in butter (add palm oil and animal fat).

14. Page 11. Line 13. ... "versus butter" should be omitted.

15. Page 13, line 24. The sentence ending in ref. (21) should be omitted. Trans fat production by hydrogenation of margarines bears no relationship to processing of edible oils.

16 Id, line 31. Cite ref. 2 here.

17. Page 14, line 24. Second colon misplaced.

Response to points 11-17: changes made

18. Id, lines 26-31. That individuals may have changed their behaviours in different ways to accommodate additional fats is a real fact of daily (eating) life, not a limitation to the study.

Response: we agree as we designed this as a pragmatic real life trial though some other reviewers have suggested this as a limitation

19. Id, lines 45-49. Which fat at a daily 50 g dose can raise LDL-C levels as shown in metabolic ward studies? Butter?

Response: we have clarified this; studies have mainly focussed on saturated fats including butter and beef fat and the reviews on different saturated fatty acids.

20. Id, sentence at bottom. Authors should discuss the concept of energy compensation via food displacement, well demonstrated in trials of other high fat foods such as nuts.

Response: we have included this in the discussion

21. Bottom page 14 to top page 15. It is unclear for which ethical reasons one must excluded patients with known prevalent cardiovascular disease, cancer or diabetes when performing a short-term nutrition intervention trial like this one. Also there is no clear reason to exclude those on lipid lowering medication. Neither prior systemic disease or lipid lowering medication are formal contraindications to a high fat diet, even less if only followed for just 4 weeks.

Response: we wished to be extra cautious in this fat supplement trial to exclude anyone at high cardiovascular risk whom it might be postulated to have deleterious effects and the ethics Committee agreed. We excluded those on lipid lowering medication as otherwise, including them would be severely criticised as compromising the primary endpoint, LDL-Cholesterol.

22. Page 15, line 20. A reference is needed for the sentence ending "clearly other pathways have been postulated". Again, please provide an appropriate reference in next par., lines 31-32.

Response: included.

23. Id. Line 22. Colon misplaced.

Reviewer: 4

Comments:

This is a randomized clinical trial conducted by a team of established investigators led by Dr. Khaw at Cambridge University.

The trial compared the effects of three oils/fats, i.e., extra virgin coconut oil, butter, or extra virgin olive oil, on some cardiometabolic risk factors or markers. The main findings are that butter significantly increased LDL levels in comparison with the other two types of oils, whereas coconut oil significantly increased HDL levels in comparison with the other oils/fats. Overall, this is a rigorously-conducted clinical trial that addressed a critical research question. I have some thoughts for this excellent work, aiming to clarify certain issues.

Major comments:

1. In the description of eligibility criteria for study participants, were they taking lipid-lowering medications or not? Reading the abstract, my impression is that all participants did not take such medications, whereas when reading the method section, I have the impression that they actually did.

Response: Taking lipid lowering medication was an exclusion criterion. We have clarified this in the methods.

2. The duration of fasting is known to influence blood lipid levels. The participants were required to fast for at least 4 hours. Have the authors considered stratifying the analysis by duration of fasting, say 4-8 hours vs ≥ 8 hours?

Response: Most of the participants fasted overnight ie more than 8 hours. 12 participants fasted 4-8 hours at baseline and 23 participants fasted 4-8 hours at the follow up; the proportions did not differ significantly between intervention groups. In addition, it is well documented that fasting status does not make a difference for total, HDL or LDL-cholesterol, but only for triglycerides.

3. It is important to understand whether samples from different intervention groups were assayed in a randomized and balanced manner across batches.

Response: Yes, the samples were assayed in a randomized and balanced manner across two batches (one batch at baseline, and a second batch at follow-up). The lab personnel were blinded to intervention group status. This information is described in the methods on page 8.

4. The current trial is an open-label study. The participants knew what they were given. Is the self-reported compliance validated using biomarkers or other measures?

Response: the compliance was self reported but not validated using biomarkers. We have stated this as a limitation in the discussion section on page 17.

5. Is the 0.16 mmol/L difference in HDL levels between coconut oil and olive oil groups after the trial the difference between delta of HDL in each group or the difference of mean HDL levels after the intervention? It's unclear in the manuscript.

Response: we have clarified this in the manuscript; it is the difference between the delta of HDL-C in each group.

6. It is interesting to observe that at study baseline participants who were randomized to coconut oil groups already had higher HDL levels (+0.2 mmol/L) than the olive oil group. Likewise, coconut oil group had lower LDL levels (-0.2 mmol/L) than olive oil group at baseline. It seems that the randomization did not generate balanced distribution of baseline blood lipid levels between the three intervention groups.

Response: It would be unlikely to have perfectly identical values following randomization but the issue is whether this would have affected the response to fats. Within the range of 0.2mmol/L it would seem implausible. Indeed it is notable that the mean LDL levels were similar in coconut oil and butter intake groups at baseline, but these changed differently over the intervention period. The outcome was comparison of the change in lipid concentrations between the different fat allocations.

7. One limitation that deserves some attention. As total energy and dietary carbohydrate intake may also influence blood lipid levels, to test independent effects of different fats, intake of carbohydrates (% of energy) should be maintained in an isocaloric setting between groups. The current trial did not seek to maintain a comparable carbohydrate intake level, nor a constant calorie intake. Such a limitation deserves some further discussions.

Response: we have discussed this further in the manuscript. There clearly is a difference of opinion; reviewer 2 did not consider this a limitation but more a reflection of a real life setting in free-living populations. Indeed, as is well acknowledged, a more “invasive” dietary intervention such as maintaining a constant calorie intake is very challenging and compromises adherence. We now show dietary data at the beginning and the end of the trial.

Reviewer: 5

Comments:

This is an interesting but (too) naive article.

To improve it, it would be important to clearly describe the full fatty acid composition of each tested fat, to measure the plasma and red cell fatty acid composition of each subject before and after intervention and then study potential association between each class (or each fatty acid) of fatty acid and the various endpoint

Response: We have now obtained and documented the fatty acid composition of the tested fats, and this was as expected (described within the manuscript). We have not as yet measured the fatty acid composition of each participant before and after the intervention which we are planning to do in the future but nevertheless we believe this trial documents in a real life situation responses to consumption of different fats.

VERSION 2 – REVIEW

REVIEWER	Qi Sun Harvard T.H. Chan School of Public Health, Boston, MA, USA
REVIEW RETURNED	20-Oct-2017

GENERAL COMMENTS	My previous comments have been addressed.
---

REVIEWER	Kiran Ahuja University of Tasmania, Australia
REVIEW RETURNED	31-Oct-2017

GENERAL COMMENTS	The authors have tried to answer all the queries and concerns raised from their previous submission to the BMJ. I have some remaining concerns - data presented in Figure 2,3 does not match to that presented in Table 2 at all places e.g. according to Table 2 - systolic BP is higher for butter than for coconut oil (+3.7 mmHg) but in figure three it is showing to favour butter. similarly diastolic is showing to be favouring coconut oil but butter seems to be lower in table 2. otherwise also, I feel that figure 2,3, 4 and 5 are repeating table 2 un-necessarily.
---

	- Although authors indicated that they also did per-protocol analyses. They have not mentioned anything in results especially whether the results for ITT and PP analyses was similar or different. - not sure why the data on education and job status is included since it has not been discussed anywhere in relation to the results or otherwise. -discussion seems long-winded with no clear conclusion (especially in the abstract). Based on the data presented, I think the authors should take a stand and say that coconut oil has fared better than butter and same as olive oil. I am sure the authors understand very well that guidelines are not changed based on one small study results - I feel that discussion is redundant in context of this particular study.
--	--

REVIEWER	Emilio Ros, MD, PhD Hospital Clínic, Barcelona, Spain
REVIEW RETURNED	03-Nov-2017

GENERAL COMMENTS	Authors have adequately addressed the major points raised in my review of the original BMJ submission and the manuscript is now much improved. A few minor aspects still need to be dealt with:  1. Intro, page 5, lines 51-53. The statement that “in the UK the overall dietary patterns are very different” needs a comparator of diets elsewhere, if prior dietary trials were conducted in the US or East Asia, for example. 2. At the end of Results, the elaborate description of fatty acid composition of the three oils/fats used in the intervention is unnecessary as this is a repetition of data in Supplementary Table. 3. The Discussion is exceedingly long. Some parts can be omitted without subtracting meaning to the arguments set forth, for instance, the whole paragraph in lines 33 to 45 in page 18. The first par. of Implications in page 19 can also be shortened, as well as the general discussion.
---

REVIEWER	Chan Yiong Huak Yong Loo Lin School of Medicine, National University of Singapore
REVIEW RETURNED	14-Nov-2017

GENERAL COMMENTS	Kindly provide the adjusted (for demographics and relevant confounders) estimates for the differences across the groups. The sample size calculations (eg mean diff = 0.45 & sd = 1.04) cannot have n = 30 with 80% power; kindly verify.
---

REVIEWER	Russell de Souza McMaster University, Hamilton, ON, Canada I have served as an external resource person to the World Health Organization’s Nutrition Guidelines Advisory Group on trans fats, saturated fats, and polyunsaturated fats. The WHO paid for my travel and accommodation to attend meetings from 2012-2017 to present and discuss this work. I have also received compensation for a lecture on dietary fat given at McMaster Pediatric Nutrition Days in 2016.
REVIEW RETURNED	05-Dec-2017

GENERAL COMMENTS

This is a well-conducted 3-arm parallel randomized controlled trial that compares the effect of 3 dietary fats (butter, coconut oil, and extra-virgin olive oil) on serum lipids, blood pressure, and measures of adiposity over a 4-weeks.

I commend the authors on a nice discussion section, which has explained several of the concerns/questions I had while reading the manuscript. I nevertheless have a few comments and points to clarify.

1. The authors note that their outcome variable of interest was "change from baseline" , but also included the baseline value of the variable of interest in the model as an independent variable. It is generally appreciated that change scores and baseline scores are correlated. The preferred approach is to either present end values, adjusted for baseline (ANCOVA); or simply change scores (Vickers, 2001) Might the authors justify why they used change scores, adjusted for baseline.

2. The authors state that they used the method of Borm et al. to adjust their sample size from the traditional t-test derived values to accommodate the ANCOVA approach. They state that the correlation between baseline and end values is 0.79. I'm not clear how this Borm "factor" was used. With 30 per group as per the t-test calculation, applying the Borm factor with an $r=0.79$ (or $r\text{-squared} = 0.62$), the required sample size per group is 12 ($30 * 1 - 0.62 = 11.3 + 1$). If the "value of 0.79" was the r-squared, this number would drop to 7. Might the authors be able to provide clarity on how the Borm adjustment was applied?

3. If electronic dietary recording was used, I wonder why dietary records were not collected at more points during the study. Could the authors please justify the decision to collect data only at the baseline and end of study?

4. I'm curious that the authors seem to have not paid careful attention to the substitution vs. addition of these fats to the diet. I understand that this was a "real-world" type of study, and it is certainly interesting to find that a mix of both approaches were used by participants, and this itself is noteworthy. However, I feel that the reported experimental conditions make it difficult to interpret the findings. A striking finding in table 3 is that the butter group increased total fat by half as much as the other two groups, but increased energy more. In fact, this group also did not change carbohydrate intake (while the other two groups decreased by very different amounts; the olive oil group decreased by twice as much as the coconut oil group). The lipid profiles don't quite follow the predicted changes (i.e. Mensink and Katan, 2003); this should be discussed. Also, I'm afraid that this makes it difficult to isolate the effect of the supplied fat on the lipid profiles-- and should be acknowledged in the limitations.

4. Was the change in carbohydrate expressed referring to "total carbohydrate" or just "available carbohydrate"?

VERSION 2 – AUTHOR RESPONSE

Reviewer: 1

Reviewer Name: Qi Sun

Institution and Country: Harvard T.H. Chan School of Public Health, Boston, MA, USA Competing

Interests: None declared.

My previous comments have been addressed.

Reviewer: 2

Reviewer Name: Kiran Ahuja

Institution and Country: University of Tasmania, Australia Competing Interests: None

The authors have tried to answer all the queries and concerns raised from their previous submission to the BMJ.

I have some remaining concerns

- data presented in Figure 2,3 does not match to that presented in Table 2 at all places e.g. according to Table 2 - systolic BP is higher for butter than for coconut oil (+3.7 mmHg) but in figure three it is showing to favour butter. similarly diastolic is showing to be favouring coconut oil but butter seems to be lower in table 2.

Response: We have corrected the data in Table 2 and figures 2 and 3

otherwise also, I feel that figure 2,3, 4 and 5 are repeating table 2 un-necessarily.

Response: We believe that the figures 2-5 are helpful in comparing the interventions diagrammatically as a summary of the results but they could be supplementary if preferred.

- Although authors indicated that they also did per-protocol analyses. They have not mentioned anything in results especially whether the results for ITT and PP analyses was similar or different.

Response: We indicate that results are reported by ITT in the manuscript. We did report that secondary analyses on the 82 participants reporting more than 75% compliance (ie PP analyses) showed similar results (page 12).

- not sure why the data on education and job status is included since it has not been discussed anywhere in relation to the results or otherwise.

Response: These data are shown to enable the reader to compare the characteristics of participants in the randomized groups

-discussion seems long-winded with no clear conclusion (especially in the abstract). Based on the data presented, I think the authors should take a stand and say that coconut oil has fared better than butter and same as olive oil.

Response: We would prefer not to make value statements such as “better” but have made a factual statement about these results in the conclusions.

I am sure the authors understand very well that guidelines are not changed based on one small study results - I feel that discussion is redundant in context of this particular study.

Response: While we entirely agree that guidelines are not changed based on one small study, we felt it was important to emphasize this point in the context of the trial being broadcast and potentially generating public interest and wanted to make the position clear.

Reviewer: 3

Reviewer Name: Emilio Ros, MD, PhD

Institution and Country: Hospital Clínic, Barcelona, Spain Competing Interests: None declared

Authors have adequately addressed the major points raised in my review of the original BMJ submission and the manuscript is now much improved. A few minor aspects still need to be dealt with:

1. Intro, page 5, lines 51-53. The statement that “in the UK the overall dietary patterns are very different” needs a comparator of diets elsewhere, if prior dietary trials were conducted in the US or East Asia, for example.

Response: We have added this comparator in relation to prior trials

2. At the end of Results, the elaborate description of fatty acid composition of the three oils/fats used in the intervention is unnecessary as this is a repetition of data in Supplementary Table.

Response: We were asked to include this by reviewers; we have now truncated this description in the text.

3. The Discussion is exceedingly long. Some parts can be omitted without subtracting meaning to the arguments set forth, for instance, the whole paragraph in lines 33 to 45 in page 18. The first par. of Implications in page 19 can also be shortened, as well as the general discussion.

Response: We have shortened the discussion.

Reviewer: 4

Reviewer Name: Chan Yiong Huak

Institution and Country: Yong Loo Lin School of Medicine, National University of Singapore

Competing Interests: No

Kindly provide the adjusted (for demographics and relevant confounders) estimates for the differences across the groups.

Response: Both the CONSORT 2010 guideline and CHMP guideline on adjustment for baseline covariates in clinical trials (EMA/CHMP/295050/2013) make it clear that baseline imbalance in covariates between randomised groups is likely to be due to chance, and post-hoc adjusted analyses can introduce bias. Following the recommendation of the CHMP guideline, in our pre-specified analysis plan we defined criteria which would be used to determine whether to perform an adjusted analysis as a sensitivity analysis, based on imbalances between groups in age, sex or BMI of a certain magnitude. These imbalances were not seen, and hence we did not perform adjusted analyses. We would prefer not to deviate from this approach by introducing post-hoc adjustments.

The sample size calculations (eg mean diff = 0.45 & sd = 1.04) cannot have n = 30 with 80% power; kindly verify.

Response: We are unsure whether the reviewer has applied the Borm adjustment when attempting to verify our sample size calculation. Please see our response to Reviewer 5 comment 2 for a detailed description of the sample size calculation.

Reviewer: 5

Reviewer Name: Russell de Souza

Institution and Country: McMaster University, Hamilton, ON, Canada Competing Interests: I have served as an external resource person to the World Health Organization's Nutrition Guidelines Advisory Group on trans fats, saturated fats, and polyunsaturated fats. The WHO paid for my travel and accommodation to attend meetings from 2012-2017 to present and discuss this work. I have also received compensation for a lecture on dietary fat given at McMaster Pediatric Nutrition Days in 2016. This is a well-conducted 3-arm parallel randomized controlled trial that compares the effect of 3 dietary fats (butter, coconut oil, and extra-virgin olive oil) on serum lipids, blood pressure, and measures of adiposity over a 4-weeks. I commend the authors on a nice discussion section, which has explained several of the concerns/questions I had while reading the manuscript. I nevertheless have a few comments and points to clarify.

1. The authors note that their outcome variable of interest was "change from baseline" , but also included the baseline value of the variable of interest in the model as an independent variable. It is generally appreciated that change scores and baseline scores are correlated. The preferred approach is to either present end values, adjusted for baseline (ANCOVA); or simply change scores (Vickers, 2001) Might the authors justify why they used change scores, adjusted for baseline.

Response: We chose to adjust for baseline (i.e. ANCOVA) rather than analyse simple change scores (unadjusted) to improve precision, as recommended in section 5.6 of the CHMP guideline on adjustment for baseline covariates in clinical trials (EMA/CHMP/295050/2013). As this document also points out, when baseline is included as a covariate, the estimated treatment effects are identical regardless of whether change from baseline or outcome at follow-up is used as the outcome variable in the model.

2. The authors state that they used the method of Borm et al. to adjust their sample size from the traditional t-test derived values to accommodate the ANCOVA approach. They state that the correlation between baseline and end values is 0.79. I'm not clear how this Borm "factor" was used. With 30 per group as per the t-test calculation, applying the Borm factor with an $r=0.79$ (or r -squared = 0.62), the required sample size per group is 12 ($30 * 1 - 0.62 = 11.3 + 1$). If the "value of 0.79" was the r -squared, this number would drop to 7. Might the authors be able to provide clarity on how the Borm adjustment was applied?

Response: 30 per group is after application of the Borm adjustment. $N=30$ per group after applying the Borm adjustment, using $r=0.79$, would imply $N=30/(1-0.79^2) = 80$ if a t-test were used without adjustment for baseline. With 30 per group, after Borm adjustment (ie. 80 per group before Borm adjustment), assuming a difference in means of 0.5 mmol/L, a standard deviation of 1.04 mmol/L, and a two-sided 2.5% significance level, the power is 78%, as shown in the Stata output below.

Response: We have clarified this in the paper and also taken the opportunity to simplify the whole paragraph so that it only discusses the primary outcome (change in LDL); we have also corrected the citation to the study from which the 0.79 correlation was obtained. We have described the power as "approximately 80%" in the revised paper, but are happy to state the actual value (78%) if the reviewer/editor prefers.

```
. power twomeans 6, diff(-0.5) sd(1.04) alpha(0.025) n1(80)
```

Estimated power for a two-sample means test

t test assuming $sd_1 = sd_2 = sd$

Ho: $m_2 = m_1$ versus Ha: $m_2 \neq m_1$

Study parameters:

alpha = 0.0250
N = 160
N1 = 80
N2 = 80
delta = -0.5000
m1 = 6.0000
m2 = 5.5000
diff = -0.5000
sd = 1.0400

Estimated power:

power = 0.7808

Please note – the first number (6) in the “power twomeans” command is arbitrary and does not affect the calculation.

3. If electronic dietary recording was used, I wonder why dietary records were not collected at more points during the study. Could the authors please justify the decision to collect data only at the baseline and end of study?

Response: The main reason was participant overload as the participants found completion of the dietary records burdensome. We did not wish to compromise the main trial by risking increased dropout.

4. I'm curious that the authors seem to have not paid careful attention to the substitution vs. addition of these fats to the diet. I understand that this was a "real-world" type of study, and it is certainly interesting to find that a mix of both approaches were used by participants, and this itself is noteworthy.

However, I feel that the reported experimental conditions make it difficult to interpret the findings. A striking finding in table 3 is that the butter group increased total fat by half as much as the other two groups, but increased energy more. In fact, this group also did not change carbohydrate intake (while the other two groups decreased by very different amounts; the olive oil group decreased by twice as much as the coconut oil group). The lipid profiles don't quite follow the predicted changes (i.e. Mensink and Katan, 2003); this should be discussed. Also, I'm afraid that this makes it difficult to isolate the effect of the supplied fat on the lipid profiles-- and should be acknowledged in the limitations.

Response: As the reviewer notes, this was designed as a “real world” pragmatic study and the main aim was that the participants consumed 50g of the allocated fat daily irrespective of what else they consumed in the context of their daily diet. We included substantial discussion about the possible explanations and limitations of the trial including this point. The reviewer states that the lipid profiles don't quite follow the predicted changes, but the inconsistent results from previous trials of coconut oil were the reason we conducted the trial and we do discuss the somewhat unexpected results extensively.

4. Was the change in carbohydrate expressed referring to "total carbohydrate" or just "available carbohydrate"?

Response: The carbohydrate intake is from available carbohydrate (obtained from the analysis (and not the subtraction method) of free sugars and complex carbohydrates and published in McCance & Widdowson's "The Composition of Foods").

VERSION 3 – REVIEW

REVIEWER	Emilio Ros Hospital Clínic, Barcelona, Spain
REVIEW RETURNED	23-Dec-2017

GENERAL COMMENTS	In their response to my qualm that the discussion was too long, with suggestions on where to shorten it, authors state that they have done so, but it doesn't seem to be true. Before revision the Discussion had 6 pages + 45 lines, now it has 6 pages plus 43 lines! Furthermore, my suggestions on where to cut down the text or shorten it have not been followed. Authors should heed my suggestions and reasonably shorten the discussion, else refute with good arguments why they don't.
---

REVIEWER	Chan Yiong Huakj Yong Loo Lin School of Medicine, Singapore
REVIEW RETURNED	18-Dec-2017

GENERAL COMMENTS	Regardless of how balance (which is looking at the univariate analysis only) the 3 groups were in their demographical/medical information, an adjusted analysis needs to be performed to handle subjects' profile and influence on the comparison of the 3 groups.
--

REVIEWER	Russell de Souza McMaster University, Ontario, Canada I have served as an external resource person to the World Health Organization's Nutrition Guidelines Advisory Group on trans fats, saturated fats, and polyunsaturated fats. The WHO paid for my travel and accommodation to attend meetings from 2012-2017 to present and discuss this work. I have also done contract research for the Canadian Institutes of Health Research's Institute of Nutrition, Metabolism, and Diabetes, Health Canada, and the World Health Organization for which I received remuneration. I have been awarded grants from the Canadian Foundation for Dietetic Research and Population Health Research Institute as a principal investigator, and am a co-investigator on several funded team grants from Canadian Institutes of Health Research.
REVIEW RETURNED	03-Jan-2018

GENERAL COMMENTS	Thank you for addressing most of my comments. I have one outstanding comment, relating to the interpretation of the findings in the discussion. I understand the "real-world" nature of the study, and again appreciate the virtue of this design. I still feel that you must acknowledge that the results cannot be cleanly interpreted as an isocaloric head-to-head comparison of butter vs. coconut oil vs. olive oil. For this, you need to have carefully controlled other components of diet-- which is how this study will be interpreted by some readers/public. I feel strongly that you must clearly state this, as the findings may be taken to support the view of one fat being "healthier" than another. I suggest the following wording, which I hope you take as a reasonable compromise:
---

	"As this was a "real-world" study, we made no attempt to control other aspects of their usual diet in particular, total energy intake. For this reason, our results cannot be taken to reflect what would happen when the only change to a diet is the substitution of one fat with another (e.g. replacing butter with coconut oil; or replacing butter with olive oil). "
--	--

VERSION 3 – AUTHOR RESPONSE

Reviewer: 3

Reviewer Name: Emilio Ros

Institution and Country: Hospital Clínic, Barcelona, Spain Competing Interests: None declared

In their response to my qualm that the discussion was too long, with suggestions on where to shorten it, authors state that they have done so, but it doesn't seem to be true. Before revision the Discussion had 6 pages + 45 lines, now it has 6 pages plus 43 lines! Furthermore, my suggestions on where to cut down the text or shorten it have not been followed. Authors should heed my suggestions and reasonably shorten the discussion, else refute with good arguments why they don't.

We did indicate in the cover letter that much of the additional material was in response to comments from reviewers such as expanding on the limitations of the study but we did not explain this in the response to reviewers. We have cut the paragraph suggested by the reviewer. However, because of the somewhat unexpected findings we wanted to ensure that all the limitations of the study were fully acknowledged make our position clear about the interpretation of the results in particular that they should not alter current dietary guidelines.

Reviewer: 4

Reviewer Name: Chan Yiong Huakj

Institution and Country: Yong Loo Lin School of Medicine, Singapore Competing Interests: No

Regardless of how balance (which is looking at the univariate analysis only) the 3 groups were in their demographical/medical information, an adjusted analysis needs to be performed to handle subjects' profile and influence on the comparison of the 3 groups.

We explained in our earlier response, why we followed guidelines for statistical analyses of trials which is not to adjust for other variables. Nevertheless, we have now conducted additional adjusted analyses to adjust for age and sex, and body mass index in the three groups and show these in supplemental table 1 and refer to this in the text. Further adjustment including smoking, alcohol intake, educational status did not materially change the results.

Reviewer: 5

Reviewer Name: Russell de Souza

Institution and Country: McMaster University, Ontario, Canada Competing Interests: I have served as an external resource person to the World Health Organization's Nutrition Guidelines Advisory Group on trans fats, saturated fats, and polyunsaturated fats. The WHO paid for my travel and accommodation to attend meetings from 2012-2017 to present and discuss this work. I have also done contract research for the Canadian Institutes of Health Research's Institute of Nutrition, Metabolism, and Diabetes, Health Canada, and the World Health Organization for which I received remuneration. I have been awarded grants from the Canadian Foundation for Dietetic Research and Population Health Research Institute as a principal investigator, and am a co-investigator on several funded team grants from Canadian Institutes of Health Research.

Thank you for addressing most of my comments.

I have one outstanding comment, relating to the interpretation of the findings in the discussion. I understand the "real-world" nature of the study, and again appreciate the virtue of this design. I still feel that you must acknowledge that the results cannot be cleanly interpreted as an isocaloric head-to-head comparison of butter vs. coconut oil vs. olive oil.

For this, you need to have carefully controlled other components of diet-- which is how this study will be interpreted by some readers/public. I feel strongly that you must clearly state this, as the findings may be taken to support the view of one fat being "healthier" than another. I suggest the following wording, which I hope you take as a reasonable compromise:

"As this was a "real-world" study, we made no attempt to control other aspects of their usual diet in particular, total energy intake. For this reason, our results cannot be taken to reflect what would happen when the only change to a diet is the substitution of one fat with another (e.g. replacing butter with coconut oil; or replacing butter with olive oil). "

Response: We have included this in the discussion.